# The impact of self-report inaccuracy in the UK Biobank and its interplay with selective participation

Tabea Schoeler ●[1,2,3] ✉, Jean-Baptiste Pingault ●[2,4] & Zoltán Kutalik ●[1,3,5] ✉

Although the use of short self-report measures is common practice in biobank initiatives, such a phenotyping strategy is inherently prone to reporting errors. To explore challenges related to self-report errors, we first derived a reporting error score in the UK Biobank (UKBB; $n$ = 73,127), capturing inconsistent self-reporting in time-invariant phenotypes across multiple measurement occasions. We then performed genome-wide scans on the reporting error score, applied downstream analyses (linkage disequilibrium score regression and Mendelian randomization) and compared its properties to the UKBB participation propensity. Finally, we improved phenotype resolution for 24 measures and inspected the changes in genomic findings. We found that reporting error was present across all 33 assessed self-report measures, with repeatability levels as low as 47% (childhood body size). Reporting error was not independent from UKBB participation, evidenced by the negative genetic correlation between the two outcomes ($r_g$ = −0.77), their shared causes (for example, education) and the loss in self-report accuracy following participation bias correction. Across all analyses, the impact of reporting error ranged from reduced power (for example, for gene discovery) to biased estimates (for example, if present in the exposure variable) and attenuation of genome-wide quantities (for example, 21% relative attenuation in SNP heritability for childhood height). Our findings highlight that both self-report accuracy and selective participation are competing biases and sources of poor reproducibility for biobank-scale research.

Genomic research is often confronted with large-scale datasets containing error in the phenotypic measures as data collection is optimized towards the recruitment of large numbers of people. To reduce participant burden, save resources and maximize sample size, recruitment schemes often favour minimal phenotyping (that is, the administration of short self-report scales) over precision phenotyping (that is, the application of gold-standard measures). In the UK Biobank (UKBB), such self-report measures serve as the primary data source for commonly studied phenotypes, notably socio-demographic data, health-related information, behavioural outcomes and lifestyles. Although all phenotypes are measured with some degree of error, including those objectively ascertained (for example, biological measures or laboratory assays), error induced by brief self-report measures pose a particular challenge when studying the associations of those phenotypes with genetic or other phenotypic information. As the reported information is influenced by subjective interpretation, misreporting or cognitive

[1]Department of Computational Biology, University of Lausanne, Lausanne, Switzerland. [2]Department of Clinical, Educational and Health Psychology, University College London, London, UK. [3]Swiss Institute of Bioinformatics, Lausanne, Switzerland. [4]Social, Genetic and Developmental Psychiatry Centre, Institute of Psychiatry, Psychology and Neuroscience, King's College London, London, UK. [5]University Center for Primary Care and Public Health, Lausanne, Switzerland. ✉e-mail: tabea.schoeler@unil.ch; zoltan.kutalik@unil.ch

**Fig. 1 | Measurement repeatability of UKBB self-report and objective measures.** $R^2$ = Variance explained by models regressing $P_{T2}$ (for example, birth weight reported at follow-up) onto $P_{T1}$ (for example, self-reported birth weight assessed at baseline), while controlling for follow-up time ($time_{T2-T1}$). Variables with $R^2$ estimates above the grey horizontal line indicate variables with high levels of repeatability ($R^2 > 0.9^2$).

limitations, error in self-report measures constitutes a potentially greater threat to the validity of findings.

While the early stages of genome-wide research were dominated by a push towards ever-growing sample sizes, challenges related to phenotype ascertainment are increasingly recognized as a non-negligible source of bias in genomic research[1,2]. Although random error in phenotypes does not lead to bias in single nucleotide polymorphism (SNP) estimates (Supplementary Fig. 1), the resulting measurement imprecision and increased type-II error rates constitute one of the causes for large sample size requirements in genomic research. If gene discovery is the primary study aim, increasing sample sizes can compensate for random error in the phenotype within the limits of feasibility. However, more problematically, random error puts an upper bound on how much variance can be explained in the phenotype[3,4]. Downstream genome-wide analyses (GWA) focusing on variance components (for example, heritability estimates[5] and polygenic prediction[6–8]) would therefore show (downward) bias in the presence of self-report inconsistencies.

Detecting and correcting self-report errors can be challenging when relying on biobank-scale data because the required validation data are rarely available. However, with the increasing availability of repeated measurements in the UKBB, it is now possible to more systematically explore causes and consequences of self-report inconsistencies across measurement occasions. In this work, we aim to contribute to the growing body of research scrutinizing the impact of study design characteristics and participant behaviour (for example, sampling procedures[9–12], missing data[13], study engagement[14] and data quality[15–17]) on findings obtained from biobank-scale data. Here we focus on the challenges related to reporting error, defined as inconsistent self-reporting across measurement occasions. To that end, we aim to quantify error in commonly studied UKBB phenotypes and assess its impact on genome-wide quantities. Further, as self-reporting represents only one among numerous participation behaviours, we also explore its interplay with a previously studied participation behaviour known to impact study findings (that is, selective UKBB participation[10,12]). Such work is not only crucial for the interpretation of findings obtained from existing biobanks but may also help shape strategies aiming to enhance phenotype resolution and recruitment strategies in future biobank initiatives.

## Results

### Indices of reporting error in the UKBB

We included 33 time-invariant self-report measures to assess inconsistencies in self-reporting. Outlier values were identified and subsequently removed for 3 of the measures, including age of 1st sexual intercourse (34 outliers removed), childhood sunburns (10 outliers removed) and age of onset of smoking (5 outliers removed). The box and scatter plots of these measures before and after outlier removal are shown in Supplementary Fig. 2. As shown in Fig. 1 (Supplementary Table 1), reporting error was present across all 33 assessed UKBB time-invariant phenotypes, with a mean error estimate of 0.21 (possible range, 0 (absence of error) to 1). High levels of measurement repeatability were present for self-reports providing information about major life events, such as date of birth ($R^2 > 0.99$), number of children ($R^2 = 0.99$) and country of birth ($R^2 = 0.99$). A substantial proportion of self-reports showed questionable levels of repeatability, notably for variables that rely heavily on recall of childhood histories, such as childhood sunburns ($R^2 = 0.53$), age at first facial hair ($R^2 = 0.50$) or comparative childhood body size ($R^2 = 0.47$). Figure 1 also shows the level of repeatability for variables containing error due to misreporting and/or temporal variability. Here, self-report measures subject to temporal instability showed particularly low levels of repeatability, notably for diet (for example, vitamin D intake in last 24 hours) and other lifestyle measures (for example, physical activity in last 24 hours). Although variations among objectively ascertained 'gold-standard' measures (for example, height and systolic blood pressure; highlighted in violet in Fig. 1) are free of error due to misreporting, measurement imprecision resulting from other sources (for example, biological fluctuations, technical challenges and data-processing errors) was nevertheless found to be non-negligible for a majority of these measures (for example, sodium concentration and hearing performance). Five UKBB phenotypes had data from directly comparable objective and subjective measures. Estimation of $R^2$ showed that the concordance between the 2 data sources (objective versus subjective) was low, ranging from $R^2 = 0.002$ (vitamin D, self-report versus blood measure) to $R^2 = 0.031$ (sleep, self reported versus accelerometer derived) to $R^2 = 0.252$ (first child's birth weight, self reported versus hospital records) (Supplementary Fig. 3 and Supplementary Table 2).

Next, we generated the reporting error scores ($RE_{sum}$; Fig. 2a), indexing the level of reporting inconsistency per phenotype and UKBB

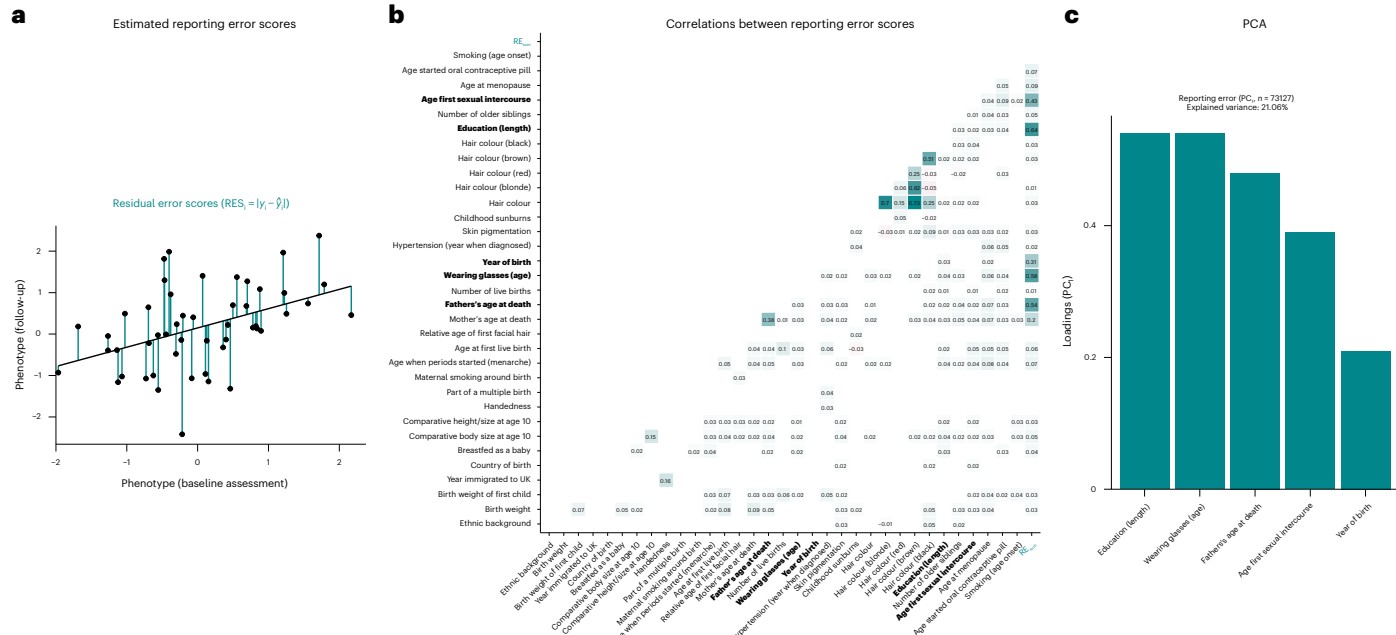

**Fig. 2 | RE_sum. a**, Illustration of a reporting error score for a particular phenotype, derived as the residual scores from a model regressing $P_{T2}$ (for example, birth weight reported at follow-up) onto $P_{T1}$ (for example, self-reported birth weight assessed at baseline). The reporting (residual) error scores are shown as the vertical deviations of the observed values ($y_i$) around the fitted line. **b**, Correlation matrix highlighting significant ($P < 0.05$, two sided) Pearson correlation coefficients between the reporting error scores. The labels in bold font highlight variables that were included in PCA. The label highlighted in turquoise (RE_sum) shows the correlations between the PCA-generated summary score (residualized for follow-up time) and the individual reporting error scores. Darker shades indicate stronger correlations. **c**, Summary of results from PCA, highlighting the variance explained by PC_1 and the loadings of the indicators on PC_1.

participant. Supplementary Figs. 4 and 5 summarize the contribution of baseline age, follow-up time, their interaction (age × follow-up time) and sex on the reporting error scores, highlighting that the scores varied mostly as a function of follow-up time and its interaction with age. In addition, reporting error was more prevalent among men, as 12 (75%) of the 16 reporting error scores showing significant sex-differential effects were higher in men than in women. The largest sex-differential effect was present for self-reported mother's age at death, where women showed substantially lower levels of reporting error.

Assessing the correlations among reporting error scores (Fig. 2b), we found that the majority of correlations were small but positive (170 (96.05%) of the 177 significant correlations). The largest positive correlations were present among measures tapping into similar constructs, such as the $r$(mother's age at death, father's age at death) = 0.38 or $r$(comparative body size at age 10, comparative height size at age 10) = 0.15. Including 5 of the reporting error scores with $n > 50,000$ in principal component analysis (PCA; years of education, age when started wearing glasses, father's age at death, age at first sexual intercourse and year of birth; Supplementary Fig. 6 for the corresponding box and scatter plots), the first principal component (PC_1) explained 21% of the variance. The individual reporting error scores all loaded positively on PC_1 (Fig. 2c). On the basis of PC_1, we computed the reporting error summary score (RE_sum), which could be generated for 73,127 individuals taking part in repeat assessments. The demographic characteristics of individuals with and without available RE_sum are shown in Supplementary Table 3.

**The link between reporting error and UKBB participation**
To examine if reporting error varied as a function of sample representativeness, we first assessed the level of covariation between reporting error and UKBB participation. Phenotypically, we found a negative correlation ($r_{Pearson} = -0.094$) between the RE_sum and UKBB participation, indicating that a greater willingness to participate in the UKBB links to more consistent self-reporting. Similarly, we observed negative genetic

correlations ($r_g$) between reporting error and other participatory behaviours, including the UKBB participation probability ($r_g = -0.77$; 95% confidence interval (CI) = -0.9 to -0.64), recontact availability in the UKBB ($r_g = -0.67$, 95% CI = -0.81 to -0.53) and follow-up (mental health survey) participation ($r_g = -0.59$, 95% CI = -0.72 to -0.45) (Fig. 4 and Supplementary Table 4).

To assess shared and non-shared characteristics between reporting error and UKBB participation, we then tested for associations between a number of baseline characteristics and the two outcomes (Fig. 3a and Supplementary Table 5). Here, significant predictors differentially linked to the two outcomes, where female participants with higher levels of education and lower body mass index (BMI) showed fewer reporting errors but a higher willingness to take part in the UKBB. Only age predicted the two outcomes in the same direction, such that older individuals tended to show more reporting errors and were also more likely to participate in the UKBB. Including all predictors simultaneously in LASSO regression explained around 12% of the variance in UKBB participation and 7% in reporting error (Fig. 3a).

The RE_sum showed low but significant levels of SNP heritability ($h^2_{RE_{sum}} = 3.19\%$, 95% CI = 1.72–4.66%). In line with the phenotypic correlations, reporting error and UKBB participation differentially correlated with most of the socio-educational and behavioural variables included in linkage disequilibrium (LD) score regression (Fig. 3b and Supplementary Table 4). These included intelligence ($r_{g_{reporting}} = -0.85$, $r_{g_{participation}} = 0.62$), years of education ($r_{g_{reporting}} = -0.81$, $r_{g_{participation}} = 0.85$) and income ($r_{g_{reporting}} = -0.70$, $r_{g_{participation}} = 0.75$). Similarly, applying Mendelian randomization (MR) analysis to identify causal factors contributing to reporting error, we find that reporting error and UKBB participation were explained mostly by socio-educational variables, where higher income, years of education and intelligence reduce self-report errors (standardized effect $\alpha_{education} = -0.34$, $\alpha_{income} = -0.33$ and $\alpha_{intelligence} = -0.27$) but increase the probability of UKBB participation ($\alpha_{education} = 0.59$, $\alpha_{income} = 0.54$ and $\alpha_{intelligence} = 0.32$) (Supplementary Table 6).

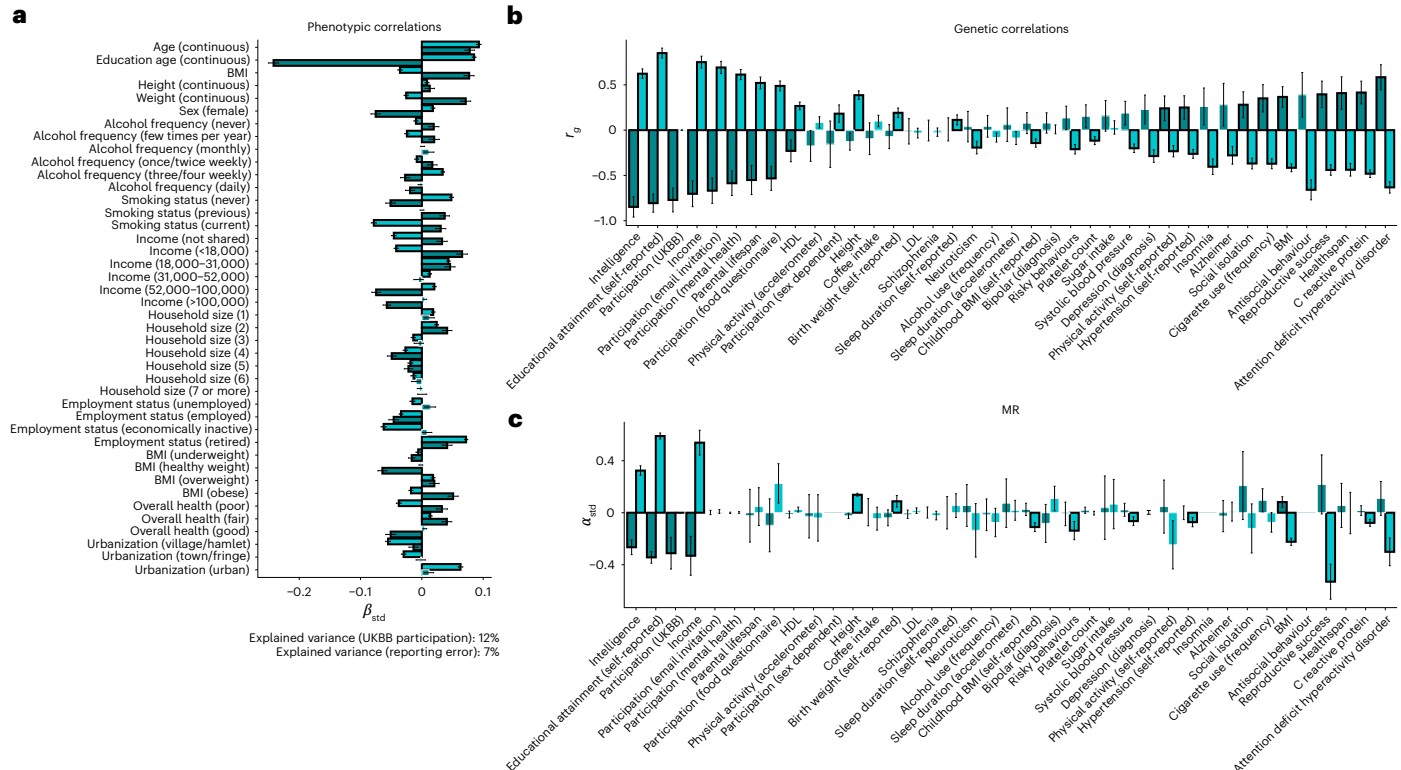

**Fig. 3 | Correlates and causes of reporting error and UKBB participation.**
**a**, $\beta_{std}$ coefficients (and 95% CIs) of variables predicting reporting error (in dark turquoise; $n = 73,127$) and UKBB participation (in light turquoise; $n = 415,066$) in univariate regression models. **b**, $r_g$ estimates and corresponding 95% CIs of reporting error ($n = 62,131$) and UKBB participation ($n = 283,749$) with other traits. Significant genetic correlations (false-discovery-rate-corrected $P(P_{FDR}) < 0.05$) are highlighted with black borders. **c**, Standardized estimates ($\alpha_{std}$, with 95% CIs) obtained from MR analyses on reporting error ($n = 62,131$) and UKBB participation ($n = 283,749$) as the outcomes. Significant MR estimates ($P_{FDR} < 0.05$) are highlighted with black borders. Two-tailed tests were used in all instances when testing for statistical significance.

Figure 4 shows the distribution of the participation (inverse probability) weights and reporting error (inverse variance) weights. The performance of the inverse variance weights was assessed in terms of reporting error reduction in 8 phenotypes, including those used in PCA and 3 additional phenotypes showing the largest degree of reporting error (that is, body size at age 10, relative age of first facial hair and number of childhood sunburns; Fig. 1). Both the inverse variance weights and the participation weights performed as intended, in that they reduced the error variance in the eight variables inspected for measurement inconsistencies (that is, increasing the level of measurement repeatability $R^2$; Fig. 4b) and made the sample more representative (that is, lowering the mean age and mean level of education; Fig. 4c), respectively. As the variability among the participation weights was large (indicating probable risk of bias due to selective participation), its application resulted in a substantial loss in effective sample size (62%, from $n = 63,896$ to $n_{effective} = 24,437$; Fig. 4a). In contrast, the reporting error weights showed little variability, causing a minimal loss in effective sample size (2%, from $n = 63,896$ to 62,636). The reporting-error-adjusted participation weights (inverse variance weights × participation weights) no longer reduced reporting error in all instances and re-introduced a slight shift towards non-representativeness, resulting in a slight increase in effective sample size compared with the unadjusted participation weights (24,437 versus 24,623).

## Simulations

We tested eight simulation scenarios to illustrate the individual and combined impact of reporting error and selective participation on exposure–outcome associations (Fig. 5). The following standardized coefficients for education ($E$) and BMI ($B$) on reporting error ($R$) and

the participation probabilities ($P$) were estimated and used to simulate (sim) the data: $R_{sim} = -0.48E + 0.03B + \varepsilon$ (with $\varepsilon$ as the error term) and

$$P_{sim} = \frac{1}{1 + \exp(-(-2.84 + 0.42E - 0.12B))}.$$

We found that deviations from the true causal effect resulted from both selective participation and reporting error in the exposure, in both cases leading to downward bias in the effect estimate (Fig. 5b). Root-mean-square error (RMSE) was most strongly increased by reporting error in the exposure (Fig. 5b), reflecting a large bias in the effect estimate towards the null. Although reporting error in the outcome did not induce bias in the effect estimate, the increased uncertainty in parameter estimates also increased the RMSE, a measure that combines both bias and variance.

## Impact of reporting error on SNP effects and trait heritability

To assess the impact of reporting error on genome-wide results, we compared the output obtained from GWA on single-measure phenotypes (for example, self-reported childhood height assessed at baseline) versus repeated-measure phenotypes (using the average across multiple measurement occasions) (Fig. 6). In total, 652 LD-independent SNPs reached significance ($P < 5 \times 10^{-8}$) in genome-wide scans on the 24 traits, of which 149 (22.85%) were only identified in repeated-measure GWA. Among the identified SNPs, the explained variance increased following error correction for 492 SNPs (75.46%). Although the $\beta$ estimates obtained from the two sets of GWA were the same (Supplementary Table 7), in accordance with the simulations demonstrating that reporting error in the outcome does not induce bias, the reduced error in the phenotype value narrowed the standard errors of the effect estimates, thereby boosting power for genome-wide discovery.

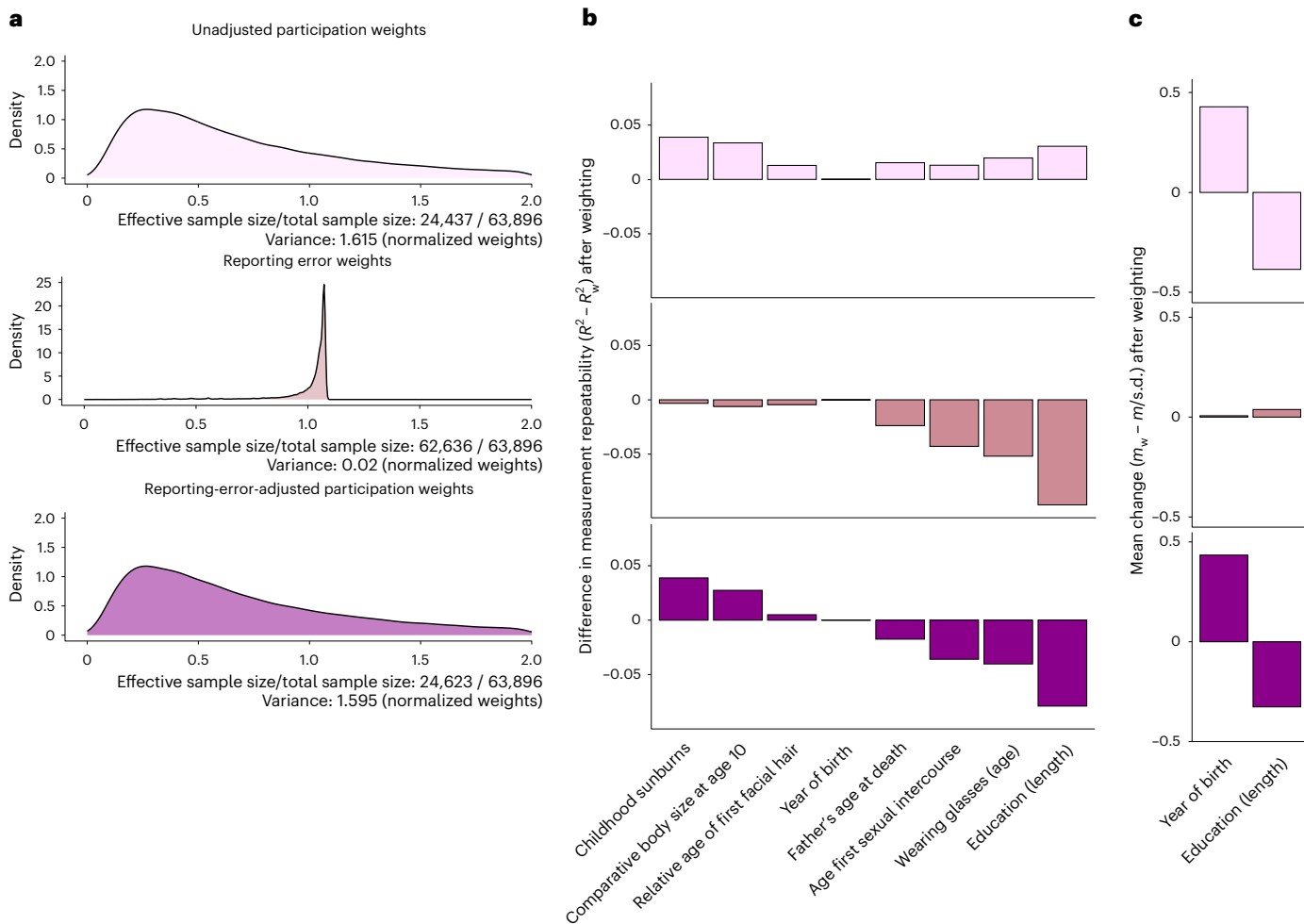

**Fig. 4 | Reporting-error-adjusted participation weights. a**, Truncated density curves of the normalized UKBB weights ($w$), estimated for $n$ = 63,898 participants. The effective sample size was estimated as $n \times \{1/[\mathrm{Var}(w)+1]\}$. **b**, $R^2$ = variance explained by standard (ordinary least squares) regression models regressing $P_{T2}$ onto $P_{T1}$, while controlling for follow-up time ($\mathrm{time}_{T2-T1}$). $R^2_w$ = variance explained by weighted (weighted least squares regression) models, incorporating UKBB weights to adjust for selective participation (top: unadjusted participation weights), reporting error (middle: reporting error weights) or both (bottom: reporting-error-adjusted participation weights). Positive values in $R^2_{\mathrm{diff}}$ ($R^2 - R^2_w$) index reduced measurement repeatability following weighting. **c**, Change in means as a function of weighting, obtained for two continuous phenotypes known to link to UKBB participation (age and education). Change in means was expressed as a standardized mean difference, that is, the difference between the unweighted mean ($m$) and the weighted mean ($m_w$), divided by the unweighted standard deviation ($m_w - m$/s.d.).

Finally, with respect to SNP-based heritability estimates, we find that enhanced phenotype resolution increased $h^2$ estimates. Overall, the degree of $h^2$ disattenuation was proportional to the degree of reporting error per phenotype ($r(h^2_{\mathrm{diff}}, R^2_{\mathrm{repeatabiliy}}) = -0.75$), with $h^2_{\mathrm{diff}} = h^2$ repeated measure $- h^2$ single measure, where the largest notable downward bias in $h^2$ estimates was present for self-reported height size at age 10 ($R^2_{\mathrm{repeatabiliy}} = 0.55$, $h^2_{\mathrm{single\sim measure}} = 23\%$ versus $h^2_{\mathrm{repeated\sim measure}} = 29\%$). The complete set of results is included in Supplementary Fig. 7 and Supplementary Tables 7 and 8.

## Discussion

Phenotyping based on short self-report measures is common practice in biobank schemes, which has paved the way for large-scale genome-wide discovery studies involving millions of individuals. Although such assessments are cost-effective and minimize the invested time of participants, they are particularly prone to errors resulting from misreporting. In this study, we quantified the extent of reporting error for commonly studied UKBB phenotypes, assessed its properties and links with other participation behaviours, and evaluated its impact on exposure–outcome and genotype–phenotype associations.

Overall, we found that reporting error is non-negligible for many commonly studied self-report measures, notably those relating to early life histories (for example, puberty, education and childhood height/weight), common environmental exposures (for example, number of sunburns) or lifestyles (for example, age when started smoking). Consequently, exploiting large biobank samples does not necessarily enhance the signal-to-noise ratios for these phenotypes, as loss of power resulting from reporting error may equate to discarding up to half the sample (as would be expected if the square of the correlation between the true phenotype and the measured phenotype is around 0.5 (ref. 18)). Considerations on statistical power and sample size requirements should therefore not only focus on the genetic architecture of the trait and the study design but also incorporate phenotype resolution as a parameter of interest.

Examining factors contributing to reporting error, we found that reporting error varied systematically across socio-demographic groups. In particular, young female participants with higher intelligence scores and those from a socio-economically favourable background (higher education and income) tended to provide the most accurate self-report information. This is consistent with the notion of heteroskedastic error, where the error variance depends on certain

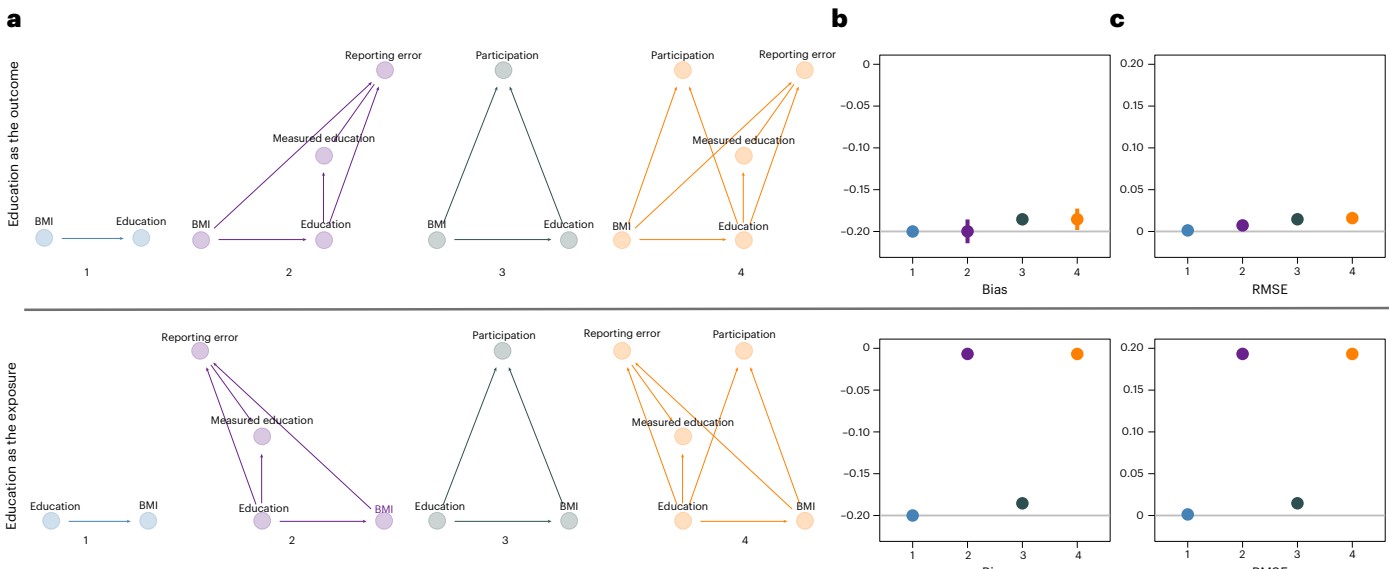

**Fig. 5 | Simulations illustrating the impact of reporting error and/or selective participation on exposure–outcome associations. a**, Directed acyclic graphs illustrating the different simulation settings, including either the ground-truth scenario (no participation bias or reporting error; 1, highlighted in blue) or scenarios where reporting error (2, highlighted in violet), participation bias (3, highlighted in green) or both (4, highlighted in orange) were present when assessing the effect of BMI on self-reported education (top) and the effect of self-reported education on BMI (bottom). **b,c**, The impact of the two participatory behaviours (reporting error and selective participation) in each of the simulated scenarios was assessed in terms of bias (**b**; 1 and 2, showing the difference between the estimated coefficient (*y* axis) and the true estimate of the exposure–outcome association (grey line, where the true causal effect was set to be −0.2)) and RMSE (**c**; 1 and 2, showing RMSE on the *y* axis, with the grey line indicating RMSE = 0) when testing the association between education and BMI. Data were simulated to mimic the UKBB response rate, where around 5.5% of the simulated data (for *n* = 9,000,000 individuals) were selected. All error bars shown in the figure represent the 95% CIs.

sample characteristics (for example, the accuracy in reporting level of education depends on education itself; Fig. 5a). The impact of this error structure on study findings will depend on the research question of interest: if gene discovery is the main goal, error in the phenotype reduces power and increases type-II error rates. Although increasing the sample size (that is, reduced sampling error) could compensate for the loss of power, such efforts would not correct for the downward bias in estimates of variance components (for example, SNP heritability and polygenic prediction) resulting from error in the phenotype. For example, for phenotypes with high levels of reporting error, we observed relative $h^2$ attenuation of up to 21% (Fig. 6a). As such, part of the missing heritability problem results from poor phenotype ascertainment, such as the use of minimal phenotyping or misclassification[1]. Similarly, the higher $h^2$ observed for physical attributes (for example, height and eye colour) than that for socio-behavioural traits (for example, smoking and socio-economic status) in the UKBB[19] may not solely reflect a stronger genetic component because measurement problems are mostly inherent to the latter traits. For polygenic scores in particular, attenuation bias due to self-report error can be twofold: first, high phenotypic error in the discovery sample increases measurement error in polygenic indices, leading to attenuation of their effects when tested in replication samples. Second, this bias is expected to be further amplified if self-report accuracy is low in the replication sample.

In classic observational analyses, bias will occur if reporting error is present in the exposure, which attenuates effect estimates towards the null (for example, regression dilution or attenuation bias[17,20]). In this scenario, the bias on parameter estimates can be particularly large, potentially exceeding bias resulting from other sources (for example, selective participation; Fig. 6b). As such, although large-scale biobanks are imperative for the study of biological pathways of small effects, such minimally phenotyped convenience samples may not be a strong contender for classic (non-genetic) epidemiological research. For that, smaller but more representative samples with gold-standard measures are potentially the more trustworthy alternative.

Finally, we compared features underlying reporting error with those of other participation behaviours, here the UKBB participation propensity. We found that individuals with high self-report quality were more likely to participate in the UKBB and that the application of statistical tools designed to ensure sample representativeness (probability weighting) increased self-report errors. This finding is consistent with findings from survey research, where probability (that is, representative) samples showed more measurement error than volunteer samples[21] and where efforts to enhance data quality reduced sample representativeness[22,23]. Together, these results highlight that biases resulting from response and participation behaviours are not independent and operate in opposite directions, such that adjusting for one type of bias could aggravate bias resulting from other sources. Consequently, design considerations should also focus on finding an optimal trade-off between sampling bias and phenotype precision. For example, the application of reporting error (inverse variance) weights enhanced phenotype resolution in the UKBB without further compromising the level of representativeness in the UKBB (Fig. 4). Collecting quality indicators and metrics for phenotype precision for all individuals when assessed at study entry (for example, use of tools to screen for poor questionnaire responding[24]) in future biobanks may therefore prove useful to remove some of the noise in the phenotype. Alternatively, researchers may choose to average phenotype scores across multiple measurement occasions if repeat-measurement data are available.

A key consideration when interpreting our results relates to the error structure examined here. More specifically, our work focused on inconsistent self-reporting over time (that is, random fluctuations in the phenotype), rather than sources of consistent misreporting (that is, systematic over- or under-reporting; Supplementary Fig. 1d). Systematic error, documented for numerous traits (for example, self-reported weight, where overweight individuals tend to under-report[25]), can only be explored if error-free reference data are available. For that reason, it was also not possible to explore error in phenotypes subject

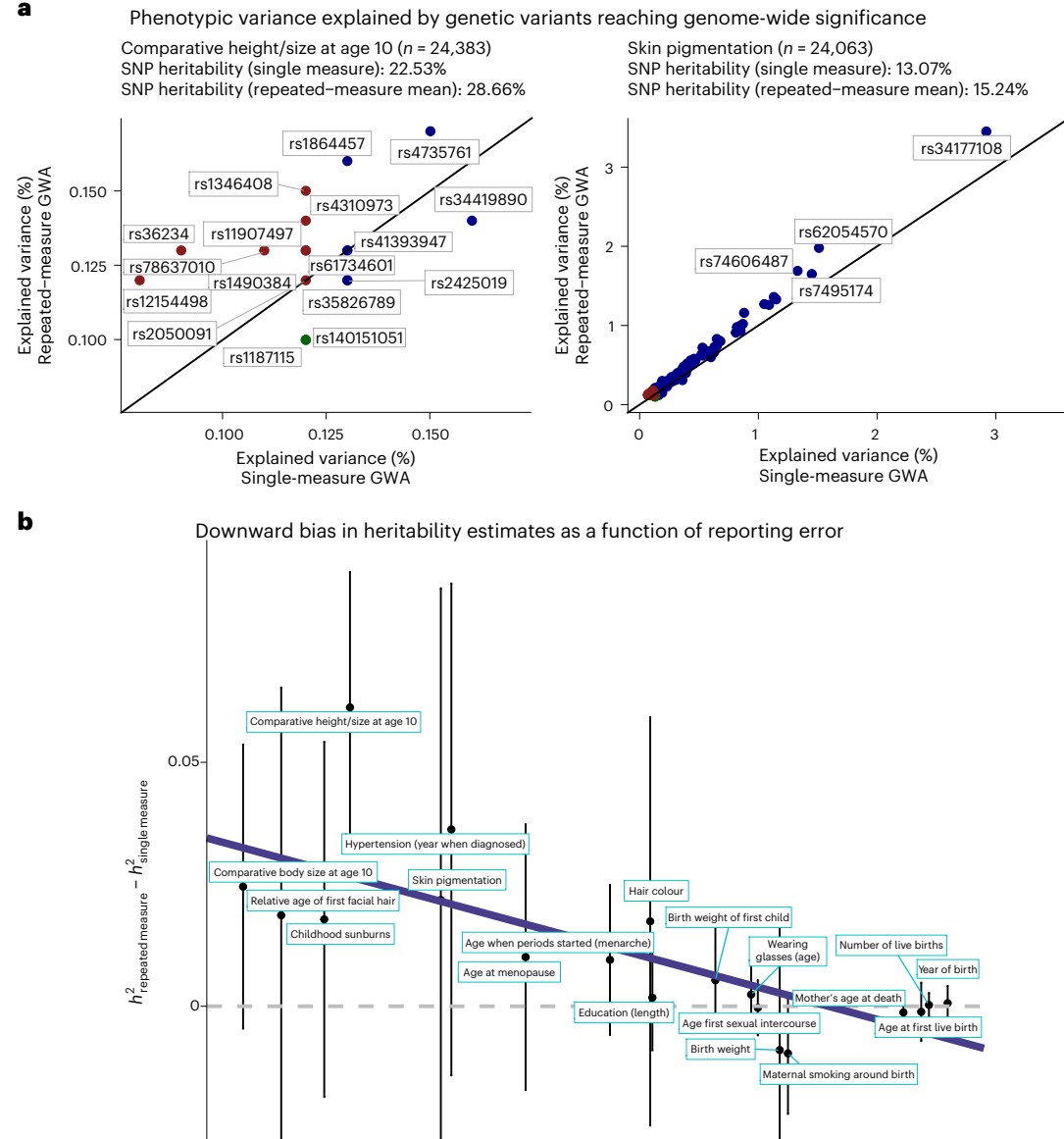

**Fig. 6 | Impact of reporting error on SNP effects and trait heritability.**
**a**, Explained variance ($\beta_{std}^2$) per SNP reaching genome-wide significance in error-corrected GWA analyses (*y* axis; phenotype obtained using means across multiple measurement occasions) or error-uncorrected GWA analyses (*x* axis; phenotype obtained from a single baseline measure). The colour scheme highlights in which GWA the genetic variant was identified, including error-corrected GWA (in red), error-uncorrected GWA (in green) or both (blue). **b**, The *y* axis shows the differences in SNP heritability estimates obtained from error-corrected GWA analyses and error-uncorrected GWA analyses ($h^2_{diff} = h^2_{repeated\,measure} - h^2_{single\,measure}$). The *x* axis gives the degree of repeatability per phenotype, estimates as the variance ($R^2$) explained by models regressing $P_{T2}$ on $P_{T1}$, while controlling for follow-up time ($time_{T2-T1}$) and age. The sample size per phenotype is included in Supplementary Table 8. All error bars shown in the figure represent the 95% CIs.

to temporal variability (for example, self-reported alcohol use) because the data at hand did not allow us to distinguish reporting error from environmental influences on the observed within-individual variability. In addition, our derived reporting error composite score reflects only an imperfect approximation of its underlying construct (reporting error propensity). Implementing strategies to enhance the resolution of this measure (for example, by using additional follow-up waves when deriving the individual reporting error scores), alongside

explorations of alternative structural models (for example, single-trait versus single-factor versus multifactor analyses to capture dimensions of reporting error) could therefore prove useful in future investigations. Finally, the reporting error mechanisms identified in this work may not translate to other cohorts because differences in recruitment schemes and population characteristics probably impact how error in self-report measures is expressed. Future research exploring how self-report error manifests in samples with different characteristics (for

example, enriched for individuals from disadvantaged backgrounds, with poorer health or from younger samples) is therefore needed to assess how different recruitment strategies may impact bias resulting from reporting error and/or sample non-representativeness.

In summary, our findings emphasize that both self-report data quality and sampling features are potential sources of poor reproducibility for biobank-scale research, leading to imprecision and bias that can complicate the interpretation of findings. Analogous to quality control procedures developed for the processing of genetic data, the application of tools designed to enhance phenotype resolution (for example, repeat measurements, regression calibration[17], imputation[26] and weighted regression) and sample representativeness (for example, probability sampling or weighting) should therefore become an integral part of data collection, pre-analytic data handling and sensitivity checks.

## Methods

### Indices of reporting error in the UKBB
This research has been conducted with the UKBB resource under application number 16389. The UKBB is a large prospective study assessing more than 500,000 participants aged between 40 and 69 years who attended 1 of the baseline assessment centres between 2006 and 2010[27]. Included in this work were individuals with at least 1 follow-up assessment, including either individuals taking part in the 1st repeat assessment centre (around 20,000 participants living within 35 km of the Stockport Biobank coordinating centre[28]) or the brain magnetic resonance imaging assessment (ongoing, inviting back up to 100,000 of the original volunteers[29]). We first screened all UKBB phenotypes that could be used as indices of reporting error, defined as inconsistent self-reporting over time. To that end, we included phenotypes that were assessed longitudinally but represented time-invariant variables, namely, those that cannot change following the baseline assessment (for example, self-reported birth weight, number of older siblings and age at first sexual intercourse). To minimize the impact of possible outlier values in continuous variables (variables with >10 levels, for example, age when started smoking), we excluded baseline and/or follow-up observations with large deviations (≥10 s.d.) from the baseline mean. For each of the included time-invariant phenotypes, we partitioned the variance of the phenotype into its error-free and reporting error component by regressing phenotype assessed at time point 2 ($P_{T2}$; for example, self-reported birth weight at follow-up) onto the phenotype assessed at time point 1 ($P_{T1}$; for example, self-reported birth weight at baseline). Follow-up time (time between $P_{T1}$ and $P_{T2}$; $time_{T2-T1}$) was included as a covariate in this model ($P_{T2} = P_{T1} + time_{T2-T1}$). The variance explained by the model ($R^2$) was used as an index of phenotype repeatability, such that $1 - R^2$ quantifies the level of reporting error per phenotype. For comparison, we also estimated $R^2$ for phenotypes subject to within-person temporal variability (including only objectively ascertained phenotypes, for example, BMI and LDL) and measures subject to both temporal variability and reporting error (for example, self-reported alcohol use and physical activity).

Next, to explore some of the properties underlying reporting error, we derived individual reporting error scores using a two-stage protocol. In stage 1, we extracted the residuals ($|RES_i|$) from a model regressing $P_{T2}$ on $P_{T1}$ (see Fig. 2a for an illustration). In stage 2, the scaled residuals ($|RES_i|/s.d._{T1,T2}$) from stage 1 were then used as input for PCA to obtain a weighted $RE_{sum}$. In PCA, we included only reporting error scores with at least 50,000 non-missing repeated observations. After combining the selected scores, we imputed missing values using row-wise mean imputation and performed PCA. On the basis of $PC_1$, we then generated the weighted summary scores ($RE_{sum}$) from the values of their observed indicator items and residualized $RE_{sum}$ for follow-up time ($time_{T2-T1}$). This score is a (weighted) average of reporting errors, representing the overall inaccuracy an individual shows when responding to time-invariant questions repeated over time. The resulting summary

scores were used as the primary outcome in downstream analyses exploring correlates and causes of reporting error. The UKBB resource was approved by the UKBB Research Ethics Committee, and all participants provided written informed consent to participate.

### GWA
The $RE_{sum}$ was then subjected to a genome-wide scan. For all GWA, we restricted the sample to individuals of European ancestry based on principal components and excluded individuals with a high missing rate (that is, proportion of genotypes not called) and/or high heterozygosity on autosomes (that is, proportion of autosomal heterozygous calls). Here, the UKBB[30,31] flagged 968 samples as outliers due to high missingness and/or extreme heterozygosity that was not explained by mixed ancestry or increased levels of marriage between close relatives. Extreme values in these metrics can be indicators of poor sample quality (for example, due to DNA contamination) and were therefore discarded. Genetic variants were filtered according to Hardy–Weinberg disequilibrium ($P > 1 \times 10^{-15}$), minor allele frequency (>1%), minor allele count (>100) and call rate (>90%). The association tests were performed in REGENIE v.3.2.6 (ref. 32), adjusting for age, sex and the first ten principal components. The resulting $RE_{sum}$ summary statistics file was then included in LD score regression[33] (as implemented in Genomic-SEM[34]) to estimate SNP heritability and genetic correlations with other traits. Genetic correlations were estimated for 39 publicly selected traits with available summary statistics files, where the selected traits tapped into participation behaviours (for example, the UKBB participation probability and recontact availability in the UKBB), physical features (for example, height and BMI), biological markers (for example, LDL and systolic blood pressure), lifestyles (for example, smoking and coffee intake), social variables (for example, socio-economic status and education) and mental health/personality (for example, schizophrenia, attention deficit hyperactivity disorder and neuroticism) (see Supplementary Table 9 for details and references). To identify causal factors contributing to reporting error, we performed MR as implemented in the R package TwoSampleMR[35]. Here, we used the same 39 selected traits with publicly available summary statistics files to extract genetic instruments for the exposure, where we selected LD-independent (–clump-kb 10,000 –clump-r2 0.001) SNPs reaching genome-wide significance ($P < 5 \times 10^{-8}$). We only performed MR for exposures with at least five genetic instruments. Tests of causality were performed using the inverse-variance-weighted MR estimator, where the reporting error GWA output was included as the outcome. To facilitate comparability of the results, we standardized the SNP effects ($\beta_{std}$) before conducting MR. $\beta_{std}$ per SNP $j$ was obtained by dividing the $Z$-score per SNP ($Z_j = \beta(SNP_j)/s.e.(SNP_j)$) by the square root of the sample size ($\beta_{std}(SNP_j) = Z_j/\sqrt{N}$). The results were corrected for multiple testing using false discovery rate correction (controlled at 5%), correcting for the total number of performed tests per downstream analysis (linkage disequilibrium score regression (LDSC) and MR).

### The link between reporting error and UKBB participation
To explore patterns of covariation between reporting error and other participatory behaviours that are known to bias genome-wide estimates, we also included 'UKBB participation probabilities' in the analytical pipeline described above. This trait was derived as part of a previous study[10] focusing on the impact of participation bias on genome-wide findings. In brief, the participation probabilities are the predicted probabilities of UKBB participation (with 1 = individuals taking part in the UKBB and 0 = individuals taking part in a representative reference sample, the Health Survey England[36]) based on 14 harmonized demographic, social and lifestyle variables. In brief, taking the inverse of the participation probabilities serves as a statistical tool to correct for bias induced by selective participation, as is commonly used in surveys[37,38], classic epidemiological studies[39,40], electronic health record studies[41,42] and, more recently, in volunteer biobank samples[10,12,43]. The probability

weights included in this work have previously been validated[10] based on external data drawn from representative samples (the Health Survey England[36] and UK Census Microdata[44]) and negative control analyses (genetic analyses on sex[45]). A more detailed summary of the validation procedures is included in Supplementary Methods. Phenotypically, we estimated the level of covariation between the $RE_{sum}$ and the UKBB participation probability. In addition, we obtained the standardized coefficients of the 14 baseline variables predicting UKBB participation (representative sample = 0; UKBB = 1), as was done in our previous work[10], to compare the coefficients to those obtained when including the $RE_{sum}$ as the outcome. The total variance explained by the 14 predictors was obtained from LASSO regression (5-fold cross-validation) in glmnet[46], which also included all possible 2-way interaction terms among the categorical (dummy) and continuous variables. To assess if UKBB participation and reporting error share similar genetic and causal structures, we applied the same genome-wide pipeline as described above (that is, performing LDSC regression and MR analyses) to UKBB participation ($n = 283,749$) as the outcome of interest. The summary statistic file from the GWA on UKBB participation is accessible via the GWAS Catalog (accession number GCST90267294).

Finally, within a regression framework, adjustment for selective participation (unequal inclusion probabilities) and reporting error (unequal error variances and heteroskedasticity) can be achieved by the implementation of weights, where over-represented or reporting-error-prone individuals are downweighted and under-represented or reporting-error-free individuals are upweighted. To assess how weighting informed by participation and/or reporting error affect phenotype and sample characteristics, we derived reporting error weights ($w_{RE}$), indexed as the inverse of the error variance ($w_{RE} = 1/(1 + \sigma^2_{RE})$). $\sigma^2_{RE}$ was obtained by taking the average of the reporting error variances ($Var_P$) across the time-invariant phenotypes ($P$) selected for PCA: $Var_P = (P_{T2} - \hat{P_{T2}})^2$, where $\hat{P_{T2}}$ are the fitted values from a model regressing the standardized phenotype assessed at follow-up ($P_{T2}$) on the standardized phenotype assessed at baseline ($P_{T1}$). We then assessed changes in sample and phenotype characteristics following inverse probability/variance weighting, where we included either the UKBB participation weights ($w_P$), the error weights ($w_{RE}$) or the error-adjusted participation weights ($w_{P \times RE} = w_P \times w_{RE}$). Change was assessed at the level of (1) measurement repeatability in time-invariant phenotypes (that is, comparing estimates of $R^2$ obtained in an unweighted versus weighted sample), and (2) means in continuous phenotypes known to link to UKBB participation (that is, comparing the weighted and unweighted means obtained for years of education and age).

### Simulations

To illustrate the individual and combined impact of reporting error and participation bias on exposure–outcome associations in a realistic setting, we simulated data for two phenotypes included in exposure–outcome linear regression models (education and BMI), the two participation behaviours of interest (reporting error and study participation) and modelled the relationships among these variables. The two phenotypes of interest, BMI and education, were chosen as these represent two continuous traits with different measurement properties (reporting-error-free versus reporting-error-prone measure, respectively) and have been linked to UKBB participation[10].

The following simulation scenarios were tested: (1) the ground truth, where the causal effect of the exposure on the outcome was estimated in a representative sample, and the exposure and outcome were measured without error; (2) reporting-error-only scenario, where reporting error was present in the exposure or outcome measure (but no participation bias); (3) participation-bias-only scenario, where we introduced participation bias (but no measurement error); and (4) a scenario in which both reporting error and participation bias were introduced. These scenarios were then simulated within a bidirectional

framework, testing the effects of (error-free) BMI on (error-prone) education and vice versa. The data-generating mechanisms are depicted in the directed acyclic graphs shown in Fig. 5.

The coefficients used in the simulation scenarios were derived as follows from the UKBB data: for UKBB participation, we used the standardized coefficients for education ($\beta_{edu}$) and BMI ($\beta_{BMI}$) on UKBB participation as estimated in MR (described above). To obtain the coefficients required to simulate reporting error in self-reported years of education, we regressed the reporting error score for education ($RES_{edu}$, as described above) onto education ($E$) and BMI ($B$) and extracted the standardized effect estimates: $RES_{edu} = \alpha_{edu}E + \alpha_{BMI}B + \varepsilon$.

The obtained coefficients were then used to simulate the data, where biases were introduced as follows: for participation bias, we first generated the simulated participation probabilities, $P_{sim} = \frac{1}{1 + \exp(-(\beta_0 + \beta_{edu}E + \beta_{BMI}B))}$, where $E$ and $B$ denote the simulated variables for years of education ($E$) and BMI ($B$), respectively. The variables were simulated as $E \sim N(0, 1)$ and $B \sim N(0, 1)$ when included as the exposure and as $E = vB + \varepsilon$ and $B = vE + \varepsilon$ when included as the outcome, where $\varepsilon \sim N(0, 1 - v^2)$ and $v$ denotes the true causal effect of the exposure on the outcome. The coefficient $\beta_0$ was set to mimic the UKBB response rate, where around 5.5% of the 9,000,000 individuals initially invited to take part were recruited in the study[27] ($\beta_0 = -\log(|1 - \frac{1}{0.055}|)$). Subjects were then assigned a random number $U$ from the uniform distribution $U \sim Uniform(0, 1)$ and were classified as either respondent ($U < P_{sim}$) or non-respondent ($U \geq P_{sim}$).

Reporting error was generated for one self-report measure (education, $E$) and was simulated as heteroskedastic error. Heteroskedasticity in this context refers to error in the measured phenotype ($E_{measured}$) that is non-constant and varies across individuals: $E_{measured} = E_{true} + \varepsilon_{edu}$, where $\varepsilon_{edu} \sim N(0, R)$. $R$ was simulated as $R_{sim} = \alpha_{edu}E + \alpha_{BMI}B + \varepsilon$, which was then scaled to have a standard deviation of 1 and values of $R > 0[R = (R_{sim} + |min(R_{sim})|)/s.d.(R_{sim})]$. BMI was modelled as an error-free measure in all simulation scenarios ($B_{measured} = B_{true}$).

The impact of reporting error and selective participation was assessed in terms of bias (that is, $\beta$ coefficients of the exposure–outcome association) and RMSE, an index that captures both the severity of the bias and the variance of the estimator: $RMSE = \sqrt{\frac{1}{k}\sum_k(\hat{v_k} - v)^2}$, where $\hat{v_k}$ is the estimated effect of the exposure–outcome association at simulation $k$ and $v$ is the true causal effect of the exposure on the outcome. We performed $k = 1,000$ simulations and true causal effect was set to $v = -0.2$.

### Impact of reporting error on SNP effects and heritability

To explore the impact of reporting error on genome-wide quantities, we compared the results from GWA tests on error-corrected versus error-prone versions of the same phenotype. We derived error-corrected phenotypes by taking the mean across multiple measurement occasions (for example, mean in self-reported childhood height) because the within-person average reduces the random error in a variable. The baseline phenotype assessed in the same subset of UKBB participants was used as the error-prone counterpart (for example, baseline self-reported childhood height). Genome-wide tests using REGENIE were then performed on both the repeated-measure and the single-measure phenotype. LD-independent SNPs reaching genome-wide significance ($P < 5 \times 10^{-8}$) were selected via clumping (clump-kb, 250; clump-r2, 0.1) and the explained variance per SNP $j$ was obtained by squaring $\beta_{std}$. We estimated SNP heritability for both the single-measure ($h^2_S$) and the repeated-measure GWA ($h^2_R$) and calculated the difference ($h^2_{diff} = h^2_R - h^2_S$) using the following test statistic:

$$Z_{h^2} = \frac{h^2_{diff}}{s.e.(h^2_{diff})}$$

$$\text{s.e.}\left(h^2_{\text{diff}}\right) = \sqrt{\text{s.e.}(h^2_R)^2 + \text{s.e.}(h^2_S)^2 - 2r \times \text{s.e.}\left(h^2_R\right)\text{s.e.}(h^2_S)}$$

The correlation coefficient $r(h^2_R, h^2_S)$ was obtained from 200-block jackknife analysis, where we split the genome into 200 equal blocks of SNPs and removed 1 block at a time to perform jackknife estimation. $h^2_{\text{diff}}$ was obtained for traits with at least 2% SNP heritability.

## Reporting summary

Further information on research design is available in the Nature Portfolio Reporting Summary linked to this article.

## Data availability

The reporting error genome-wide association statistics are available through the GWAS Catalog (accession number GCST90448966).

## Code availability

The following software was used to run the analyses: REGENIE (https://github.com/rgcgithub/regenie), TwoSampleMR (https://mrcieu.github.io/TwoSampleMR/) and GenomicSEM (https://github.com/GenomicSEM/GenomicSEM). All analytical scripts are available at https://github.com/TabeaSchoeler/TS2023_repErrorUKBB.

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

## Acknowledgements

We thank all biobank participants for sharing their data. This study would not have been possible without the use of publicly available genome-wide summary data and software tools. We acknowledge these resources and thank the research participants, the research teams and institutions that have contributed to this research. Computations have been performed on the HPC cluster of the Lausanne University Hospital. Z.K. was funded by the Swiss National Science Foundation (number 310030-189147). T.S. is funded by a Wellcome Trust Sir Henry Wellcome fellowship (grant 218641/Z/19/Z). J.-B.P. has received funding from the European Research Council under the European Union's Horizon 2020 research and innovation programme (grant agreement number 863981). The funders had no role in study design, data collection and analysis, decision to publish or preparation of the manuscript.

## Author contributions

Z.K. and T.S. conceptualized the study. T.S. performed the statistical analyses. Z.K., J.-B.P. and T.S. discussed the results and provided comments on the article. All authors critically reviewed the article.

## Competing interests

The authors declare no competing interests.

## Additional information

**Correspondence and requests for materials** should be addressed to Tabea Schoeler or Zoltán Kutalik.

# Reporting Summary

## Statistics

For all statistical analyses, confirm that the following items are present in the figure legend, table legend, main text, or Methods section.

| n/a | Confirmed | |
|---|---|---|
| ☐ | ☒ | The exact sample size (*n*) for each experimental group/condition, given as a discrete number and unit of measurement |
| ☐ | ☒ | A statement on whether measurements were taken from distinct samples or whether the same sample was measured repeatedly |
| ☐ | ☒ | The statistical test(s) used AND whether they are one- or two-sided<br>*Only common tests should be described solely by name; describe more complex techniques in the Methods section.* |
| ☐ | ☒ | A description of all covariates tested |
| ☐ | ☒ | A description of any assumptions or corrections, such as tests of normality and adjustment for multiple comparisons |
| ☐ | ☒ | A full description of the statistical parameters including central tendency (e.g. means) or other basic estimates (e.g. regression coefficient) AND variation (e.g. standard deviation) or associated estimates of uncertainty (e.g. confidence intervals) |
| ☐ | ☒ | For null hypothesis testing, the test statistic (e.g. *F*, *t*, *r*) with confidence intervals, effect sizes, degrees of freedom and *P* value noted<br>*Give P values as exact values whenever suitable.* |
| ☒ | ☐ | For Bayesian analysis, information on the choice of priors and Markov chain Monte Carlo settings |
| ☒ | ☐ | For hierarchical and complex designs, identification of the appropriate level for tests and full reporting of outcomes |
| ☐ | ☒ | Estimates of effect sizes (e.g. Cohen's *d*, Pearson's *r*), indicating how they were calculated |

*Our web collection on statistics for biologists contains articles on many of the points above.*

## Software and code

Policy information about availability of computer code

| | |
|---|---|
| Data collection | This research has been conducted with the UK Biobank Resource under application number 16389. |
| Data analysis | The following software was used to run the analyses: REGENIE (https://github.com/rgcgithub/regenie) version 3.2.6; TwoSampleMR (https://mrcieu.github.io/TwoSampleMR/) version 0.5.7; GenomicSEM (https://github.com/GenomicSEM/GenomicSEM) version 0.0.5. All analytical scripts are available at https://github.com/TabeaSchoeler/TS2023_repErrorUKBB. |

For manuscripts utilizing custom algorithms or software that are central to the research but not yet described in published literature, software must be made available to editors and reviewers. We strongly encourage code deposition in a community repository (e.g. GitHub). See the Nature Portfolio guidelines for submitting code & software for further information.

## Data

Policy information about availability of data

All manuscripts must include a data availability statement. This statement should provide the following information, where applicable:
- Accession codes, unique identifiers, or web links for publicly available datasets
- A description of any restrictions on data availability
- For clinical datasets or third party data, please ensure that the statement adheres to our policy

The reporting error genome-wide association statistics are available through the GWAS catalog.

# Research involving human participants, their data, or biological material

Policy information about studies with human participants or human data. See also policy information about sex, gender (identity/presentation), and sexual orientation and race, ethnicity and racism.

| | |
|---|---|
| Reporting on sex and gender | We used self-reported sex (biological attribute) in our study. |
| Reporting on race, ethnicity, or other socially relevant groupings | We included self-reported ethnic background in our descriptive analyses (UK Biobank ID: 21000). |
| Population characteristics | In genome-wide analyses, we included UK Biobank participants of European ancestry passing standard GWA analysis quality control measures. All analyses were adjusted for batch, principal components (PC1-PC5), age and sex. Exclusions during QC process (phenotypic and genetic) are detailed in the Methods. |
| Recruitment | The UK Biobank (UKBB) is a prospective population-based research resource focusing on the role of genetic, environmental and lifestyle factors in health outcomes in middle age and later life. More than 9,000,000 men and women between 40 and 69 registered with the UK NHS were invited to take part. Of those, 5.4% (~500,000 individuals) were recruited in 22 assessment centres across England, Wales and Scotland between 2006 and 2010. |
| Ethics oversight | The UK Biobank resource was approved by the UK Biobank Research Ethics Committee and all participants provided written informed consent to participate. |

Note that full information on the approval of the study protocol must also be provided in the manuscript.

# Field-specific reporting

Please select the one below that is the best fit for your research. If you are not sure, read the appropriate sections before making your selection.

☒ Life sciences ☐ Behavioural & social sciences ☐ Ecological, evolutionary & environmental sciences

For a reference copy of the document with all sections, see nature.com/documents/nr-reporting-summary-flat.pdf

# Life sciences study design

All studies must disclose on these points even when the disclosure is negative.

| | |
|---|---|
| Sample size | We derived our outcome of interest (reporting error score) for n=73,127 UK Biobank participants. |
| Data exclusions | We removed individuals with missing data in repeated measurements. |
| Replication | We did not select an independent replication sample. There are no genotype datasets of similar size in the UK for which our outcome measure could be derived, making replication currently not feasible. |
| Randomization | Not applicable |
| Blinding | Not applicable |

# Reporting for specific materials, systems and methods

We require information from authors about some types of materials, experimental systems and methods used in many studies. Here, indicate whether each material, system or method listed is relevant to your study. If you are not sure if a list item applies to your research, read the appropriate section before selecting a response.

## Materials & experimental systems

| n/a | Involved in the study |
|---|---|
| ☒ | ☐ Antibodies |
| ☒ | ☐ Eukaryotic cell lines |
| ☒ | ☐ Palaeontology and archaeology |
| ☒ | ☐ Animals and other organisms |
| ☒ | ☐ Clinical data |
| ☒ | ☐ Dual use research of concern |
| ☒ | ☐ Plants |

## Methods

| n/a | Involved in the study |
|---|---|
| ☒ | ☐ ChIP-seq |
| ☒ | ☐ Flow cytometry |
| ☒ | ☐ MRI-based neuroimaging |

## Plants

Seed stocks

Not applicable

Novel plant genotypes

Not applicable

Authentication

Not applicable

