## [Peer Review File · Nature Human Behaviour]

The impact of self-report inaccuracy in the UK Biobank and its interplay with selective participation

Corresponding Author: Dr Tabea Schoeler

Version 0:

Decision Letter:

30 April 2024

Dear Dr Schoeler,

Thank you once again for your manuscript, entitled "Self-report inaccuracy in the UK Biobank: Impact on inference and interplay with selective participation". I can't apologize enough for the unconscionable delay in communicating a decision on your manuscript and thank you for your patience.

Your manuscript has now been evaluated by 3 reviewers, whose comments are included at the end of this letter. Although the reviewers find your work to be of interest, they also raise some important concerns. We are very interested in the possibility of publishing your study in Nature Human Behaviour, but would like to consider your response to these concerns in the form of a revised manuscript before we make a decision on publication.

Additionally, your revised manuscript must comply fully with our editorial policies and formatting requirements. Failure to do so will result in your manuscript being returned to you, which will delay its consideration. To assist you in this process, I have attached a checklist that lists all of our requirements. If you have any questions about any of our policies or formatting, please don't hesitate to contact me.

In sum, we invite you to revise your manuscript taking into account all reviewer and editor comments. We are committed to providing a fair and constructive peer-review process. Do not hesitate to contact us if there are specific requests from the reviewers that you believe are technically impossible or unlikely to yield a meaningful outcome.

We hope to receive your revised manuscript within four months. I would be grateful if you could contact us as soon as possible if you foresee difficulties with meeting this target resubmission date.

- Include a "Response to the editors and reviewers" document detailing, point-by-point, how you addressed each editor and referee comment. If no action was taken to address a point, you must provide a compelling argument. When formatting this document, please respond to each reviewer comment individually, including the full text of the reviewer comment verbatim followed by your response to the individual point. This response will be used by the editors to evaluate your revision and sent back to the reviewers along with the revised manuscript.
- Highlight all changes made to your manuscript or provide us with a version that tracks changes.

Link Redacted

We look forward to seeing the revised manuscript and thank you for the opportunity to review your work. Please do not hesitate to contact me if you have any questions or would like to discuss these revisions further.

[redacted]

Reviewer expertise:

Reviewer #1: Biostatistics, genetics/genomics

Reviewer #2: Epidemiology/public health

Reviewer #3: Genomics, genetic epidemiology

REVIEWER COMMENTS:

Reviewer #1:

Remarks to the Author:

Schoeler and colleagues investigated potential self-report inaccuracy, contributing factors, and potential consequential biases. The research question is interesting and relevant to the debate on phenotyping strategy in large biobank studies. However, there are a few issues with the methods and results that may be questionable.

1. The measurement repeatability was very high for most of the 'invariable' traits. The only exception was 'childhood sunburns' (Figure 1 and row 26 in sTable 2). I have experience working on this phenotype, and I know that the low repeatability was due to three outlying subjects who reported over 100 occasions of sunburn. It is clear that these participants misunderstood or provided false answers. By removing these three data points, the measurement repeatability would increase from 10% to nearly 40%. The low repeatability is clearly a result of a lack of data quality control before analysis, rather than recall bias. Additionally, the presence of outlying values may have a strong confounding effect on their analyses of genome-wide significant variants, especially for traits that do not have a normal distribution. I encourage the authors to double-check the scatter plots of all variables to ensure that the SEsums are representative of repeatability rather than data quality.
2. The genetic correlations of reporting error between traits were generally very low, indicating that misreporting one trait, such as birthweight, would have minimal correlation with misreporting other traits. This is not only counter-intuitive but also contradictory to the assumption of the subsequent PCA and the authors' conclusion regarding participants who tend to report more accurately. In their conclusion, the authors suggest that "young, female participants with higher intelligence scores and those from a socio-economic favorable background (higher education and income) tended to provide the most accurate self-report information" (on the first page of the discussion). However, they have shown that there is no systematic reporting error across all phenotypes. The low correlations may be attributed, in part, to the data quality issue mentioned in my previous comment. The authors may also need to reconsider how to interpret the low correlation of repeatability between traits.
3. There were some typos throughout the document. For example, on the first page of the introduction, reference 34 should be 3 and 4.

Reviewer #2:

Remarks to the Author:

Given the attention and priority given to the large databanks such as UKBB that is this paper's focus the analyses presented in this carefully laid out paper are of considerable importance. There is too little attention to basic principles of epidemiological research such as reliability and validity. This paper is a refreshing deep dive into this area, and its findings are important. The authors lay out these results and suggest appropriate caution, and potential solutions for the challenges they have surfaced. They rightly highlight that it is not only self report that is subject to error, but the paper focuses on this aspect of UKB, using what should be time invariant variables across time points. In general the paper is written very well and the authors have tried to help the reader understand the sequences of complex analyses and lead them through these in a way that minimises the changes of getting lost. The finding that application of tools designed to increase representativeness of the data may increase bias related to Self report measures is very important. The presentation of findings in relation to the nature of biases at different analytical stages is very useful.

There are some suggestions for further improvement of the manuscript as follows:

There should be a clear description (I don't think I missed it) of who is in which analyses, and what are the characteristics of those who are not. The numbers mentioned are not entirely clear - 500,000 goes to around 70,000. It seems that only T1 and T2 are used, rather than looking at many fluctuations - if this is wrong then the sequence of interview timings needs to be laid out more clearly. The variables of BMI and LDL are treated for these analyses as error free (again if I've understood correctly) but they do have error too. There is a mention of gold standards - however all these types of data have challenges - to see medical records or a single time point 'precision phenotyping' as some sort of gold standard is also not correct, as anyone who has looked at these in detail will have experienced (reliability, validity - what is validity really for some of these variables?). These are all messy attempts at some underlying 'truth', but with very different errors and biases.

The use of pre-existing participation bias findings is not explained for readers who are not familiar with the earlier work. Some further description would help, including how robust this work is.

Exclusion of individuals with high missing variables and (or?) high autosomal heterozygosity needs to be better explained. Figure 5 needs more explanation.

Reviewer #3:

Remarks to the Author:

The authors present a comprehensive analysis of an important issue regarding the validity of biobank data, using the UK Biobank and simulations. There is increasing interest in and recognition of the possibility that design elements and assessments in large-scale biobanks may compromise the interpretation of studies leveraging such biobanks. Prior work has shown that participation

bias in volunteer biobanks, a growing resource for genomic and epidemiologic research, can lead to biased effect estimates. Here, the authors address the possibility that inconsistent reporting of phenotypic data may induce additional biases. They capitalize on repeated measurements of presumably time-invariant variables (e.g. childhood history of exposures, year of birth, ethnic background) and show that there is apparent reporting error in each and that reporting error and participation biases confer competing bias on effect estimates. Also of note, reporting error bias can substantially decrease effective sample size for biobank analyses.

A few issues deserve clarification:

1. Could the authors comment on how the variability of in the repeatability of time-invariant variables (with some e.g. childhood sunburns showing particularly high levels of putative error) might affect the PCA approach used to identify an overall reporting error score?
2. It might be worth highlighting the implications of the MR results for studies (e.g. All of Us) that focus on under-represented groups –ie the finding that higher SES is associated with fewer reporting errors and great participation probability. This would presumably induce even greater bias in results for lower SES (and perhaps other groups), suggesting a trade-off between reducing reporting bias and inclusion of more diverse samples.
3. I found it slightly confusing that, after detailed analyses to derive reporting error adjustments, the analysis of SNP weights simply compared GWAS using a baseline measure of a phenotype to GWAS using the average of the two time points. They demonstrate that using the average of measures results in improved power by reducing standard errors of the phenotype. Does this imply that simply averaging repeated measures can be a reasonable approach to minimizing the effect of reporting error in GWAS? Since we don't know the ground truth when dealing with discordant repeated measurements, averaging (at least for quantitative exposures or phenotypes) is straightforward.
4. The authors make glancing reference to polygenic prediction, but given the widespread use of biobank data for just this purpose and the downward bias of heritability with reporting error, it might be worth a sentence or two more on the implications of this work for biobank PGS research, including the likely variation in reporting error between discovery and validation datasets used for PGS.
5. Figures:
 - a. Fig 4c: are the 3 panels meant to mirror those in 4b? ie participation, reporting error, and both? Could clarify in legend.
 - b. Fig 5: Given structure of the DAGs, does this imply that controlling for (ie conditioning on) factors correlated with participation or reporting could introduce bias (as they are colliders)?

Overall, this manuscript provides an innovative series of analyses that have valuable implications for biobank research and can inform the design and interpretation of biobank genomic and epidemiologic analyses.

Version 1:

Decision Letter:

Our ref: NATHUMBEHAV-23103352A

31st August 2024

Dear Dr. Schoeler,

Thank you for submitting your revised manuscript "Self-report inaccuracy in the UK Biobank: Impact on inference and interplay with selective participation" (NATHUMBEHAV-23103352A). It has now been seen by the original referees and their comments are below. As you can see, the reviewers find that the paper has improved in revision (please note that Reviewer 2 only provided confidential comments to the editors). We will therefore be happy in principle to publish it in Nature Human Behaviour, pending minor revisions to satisfy the referees' final requests and to comply with our editorial and formatting guidelines.

We are now performing detailed checks on your paper and will send you a checklist detailing our editorial and formatting requirements within two weeks. Please do not upload the final materials and make any revisions until you receive this additional information from us.

[redacted]

Reviewer #1 (Remarks to the Author):

The authors have addressed all my concerns. Their responses are thorough and well-organised. Their additional analyses and findings are consistent with their initial results.

Reviewer #3 (Remarks to the Author):

The authors have revised the manuscript and included additional robustness analyses that adequately address the points raised in the review. I have no further questions and comments to address.

Version 2:

Decision Letter:

Dear Dr Schoeler,

We are pleased to inform you that your Article "The impact of self-report inaccuracy in the UK Biobank and its interplay with selective participation", has now been accepted for publication in *Nature Human Behaviour*.

Please note that *Nature Human Behaviour* is a Transformative Journal (TJ). Authors may publish their research with us through the traditional subscription access route or make their paper immediately open access through payment of an article-processing charge (APC). Authors will not be required to make a final decision about access to their article until it has been accepted. [Find out more about Transformative Journals](https://www.springernature.com/gp/open-research/transformative-journals)

Authors may need to take specific actions to achieve [compliance with funder and institutional open access mandates](https://www.springernature.com/gp/open-research/funding/policy-compliance-faqs). If your research is supported by a funder that requires immediate open access (e.g. according to [Plan S principles](https://www.springernature.com/gp/open-research/plan-s-compliance)) then you should select the gold OA route, and we will direct you to the compliant route where possible. For authors selecting the subscription publication route, the journal's standard licensing terms will need to be accepted, including [self-archiving policies](https://www.springernature.com/gp/open-research/policies/journal-policies). Those licensing terms will supersede any other terms that the author or any third party may assert apply to any version of the manuscript.

We welcome the submission of potential cover material (including a short caption of around 40 words) related to your manuscript; suggestions should be sent to *Nature Human Behaviour* as electronic files (the image should be 300 dpi at 210 x 297 mm in either TIFF or JPEG format). Please note that such pictures should be selected more for their aesthetic appeal than for their scientific content, and that colour images work better than black and white or grayscale images. Please do not try to design a cover with the *Nature Human Behaviour* logo etc., and please do not submit composites of images related to your work. I am sure you will understand that we cannot make any promise as to whether any of your suggestions might be selected for the cover of the journal.

[redacted]

P.S. Click on the following link if you would like to recommend Nature Human Behaviour to your librarian
<http://www.nature.com/subscriptions/recommend.html#forms>

** Visit the Springer Nature Editorial and Publishing website at http://editorial-jobs.springernature.com?utm_source=ejp_NHumB_email&utm_medium=ejp_NHumB_email&utm_campaign=ejp_NHumB for more information about our career opportunities. If you have any questions please click [here](mailto:editorial.publishing.jobs@springernature.com).

We thank the reviewers for their thoughtful feedback on our work. We have updated our manuscript in accordance with their comments and provide below our point-by-point responses.

Before we enumerate our responses to specific comments, we wish to provide all reviewers with a general overview of some of the changes made to our manuscript. Given the time elapsed between the initial submission of the manuscript and the receipt of the reviewer feedback (6 months), we updated the analytical pipeline to reflect the best practises available at the time of the revision of the manuscript (June 2024). First, we replaced the version 2 of the imputed UK Biobank genotype file with the version 3 imputed data due to imputation problems documented for version 2 by the UKBB (c.f., <https://biobank.ctsu.ox.ac.uk/ukb/label.cgi?id=100319> for details). Second, we updated REGENIE version 2.0.2 to 3.2.6 for genome-wide testing. Here, we also switched from analyzing multiple traits in parallel (which substantially reduces computational burden, but at the cost of precision in the estimates) to analyzing phenotypes separately in REGENIE (i.e., the more robust estimator). As these updates did not affect the interpretation of the results in any way, we have updated all genome-wide results in the manuscript and will upload the corrected summary statistic files to the GWAS-catalog upon the publication of this work.

Reviewer 1

Schoeler and colleagues investigated potential self-report inaccuracy, contributing factors, and potential consequential biases. The research question is interesting and relevant to the debate on phenotyping strategy in large biobank studies. However, there are a few issues with the methods and results that may be questionable.

R1.1. The measurement repeatability was very high for most of the 'invariable' traits. The only exception was 'childhood sunburns' (Figure 1 and row 26 in sTable 2). I have experience working on this phenotype, and I know that the low repeatability was due to three outlying subjects who reported over 100 occasions of sunburn. It is clear that these participants misunderstood or provided false answers. By removing these three data points, the measurement repeatability would increase from 10% to nearly 40%. The low repeatability is clearly a result of a lack of data quality control before analysis, rather than recall bias. Additionally, the presence of outlying values may have a strong confounding effect on their analyses of genome-wide significant variants, especially for traits that do not have a normal distribution. I encourage the authors to double-check the scatter plots of all variables to ensure that the SEsums are representative of repeatability rather than data quality.

We thank the reviewer for this important observation. It is true that we had not applied specific filters to remove outliers due to a participants' misunderstanding/misresponding as these response behaviours are, by definition, part of the very phenotype we aimed to study (i.e., self-report errors). However, we agree that extreme deviations can artificially bring down the measurement repeatability that we described for the 33 time-invariant self-report measures in Figure 1. We therefore screened all continuous variables and flagged individuals with self-report values greater than 10 standard deviations from the (baseline) mean. As show in the table below, three variables contained outlier values:

Phenotype	Assessment wave	minimum (before outlier removal)	maximum (before outlier removal)	mean (before outlier removal)	minimum (after outlier removal)	maximum (after outlier removal)	mean (after outlier removal)	N with repeat measurement	N outliers removed
Age first sexual intercourse	0	3	66	19	3	57	19	65157	34
	1	3	79	19	3	58	19		
Childhood sunburns	0	0	999	2	0	50	2	18382	10
	1	0	100	2	0	50	2		
Smoking (age onset)	0	5	69	17	5	58	17	18151	5
	1	5	65	17	5	60	17		

The table lists three variables for which outlier values were detected. For each variable, we report the mean, minimum and maximum value in that phenotype before and after the removal of the outlier value. This information is provided for the phenotype assessed at baseline (assessment wave = 0) and the phenotype assessed at follow up separately (assessment wave = 1). The last row lists the number of excluded outliers.

Following the removal of individuals with outlying values, the R^2 estimates (level of measurement repeatability) changed as follows: childhood sunburn (from $R^2=0.10$ to $R^2=0.53$ after removing 10 outliers), age at first sexual intercourse (from $R^2=0.80$ to $R^2=0.86$ after removing 34 outliers) and age at onset of smoking (from $R^2=0.47$ to $R^2=0.61$ after removing 5 outliers). As a consequence, we have now updated Figure 1 and the result section in the text. This correction did, however, not change the initial conclusions drawn from the results [e.g., Discussion: “Overall, we found that reporting error is non-negligible for many commonly studied self-report measures, notably those relating to early life histories (e.g., puberty, education, childhood height/weight), common environmental exposures (e.g., number of sunburns) or lifestyles (e.g., age when started smoking)], nor the downstream analysis results.

[Manuscript, Methods]: “To minimize the impact of possible outlier values in continuous variables (variables with > 10 levels, e.g., age when started smoking), we excluded baseline and/or follow-up observations with large deviations (≥ 10 standard deviations) from the baseline mean.”

[Manuscript, Results]: “We included 33 time-invariant self-report measures to assess inconsistencies in self-reporting. Outlier values were identified and subsequently removed for 3 of the measures, including age of first sexual intercourse (34 outliers removed), childhood sunburns (10 outliers removed) and age of onset of smoking (5 outliers removed). The box and scatter plots of these measures prior and after outlier removal are shown in sFigure 2 (Supplement).”

sFigure 2. Box and scatter plots for self-report measures containing outlier values

Panel A1-A3 display the box and scatter plots of the self-report measure assessed at baseline (P_0) and follow-up (P_1) prior to outlier removal. Panel B1-B3 display the same information after the removal of outlier values.

[Manuscript, Results]: “A substantial proportion of self-reports showed questionable levels of repeatability, notably variables relying heavily on recall of childhood histories, such as childhood sunburns ($R^2=0.53$), age at first facial hair ($R^2=0.50$) or comparative childhood body size ($R^2=0.47$).

Updated Figure 1:

Finally, we also generated boxplots and scatterplots of the self-report measures included in the reporting summary score, which is now added in the Supplement.

[Manuscript, Results]: “Including five of the reporting error scores with $n > 50,000$ in principal component analysis (years of education, age when started wearing glasses, father’s age at death, age at first sexual intercourse, year of birth, *c.f.*, sFigure 4 for the corresponding box and scatter plots), the first principal component (PC_1) explained 21% of the variance.”

sFigure 4. Box and scatter plots for self-report measures included in Principal Component Analysis

The figure displays the box and scatter plots of the self-report measures used in Principal Component Analysis, including the phenotype assessed at baseline (P_0) and follow-up (P_1)

Next, we removed the above listed outliers, re-generated the reporting error summary scores and re-ran genome-wide tests and all downstream analyses. In brief, while the updated reporting error summary score became slightly more predictive at the phenotypic level (e.g., 6% versus 7% of variance explained by demographic variables, Figure 2A) and the genetic level (e.g., SNP-heritability estimate in the initial manuscript=2.63% versus updated SNP-heritability estimate=3.19%), none of the downstream analyses led to different interpretations from the initial findings. As exemplified below based on Mendelian Randomization results predicting UK Biobank participation versus reporting error, we obtained the same conclusions after the implementation of additional quality control steps (further described in response to the next comment), highlighting that our initial set of results was not an artifact of poor data quality (please note that the order of the x-axis labels has changed):

We have now updated the manuscript and figures to accommodate the results obtained from the corrected reporting error summary score. This resulted in slight modifications when highlighting the results in the text, for example:

[Manuscript, Results]: [...] reporting error and UKBB participation differentially correlated with most of the socio-educational and behavioural variables included in LD score regression (Figure 3B, sTable 4). These included intelligence ($rg_{\text{Reporting}} = -0.9 -0.84$, $rg_{\text{Participation}} = 0.62$), years of education ($rg_{\text{Reporting}} = -0.87 -0.81$, $rg_{\text{Participation}} = 0.85$) and income ($rg_{\text{Reporting}} = -0.76 -0.70$, $rg_{\text{Participation}} = 0.75$). Similarly, applying Mendelian Randomization analysis to identify causal factors contributing to reporting error, we find that reporting error and UKBB participation were explained by mostly socio-educational variables, where higher income, years of education and intelligence reduce self-report errors (standardized effect $\alpha_{\text{Income}} = -0.36 -0.33$, $\alpha_{\text{Education}} = -0.33 -0.34$, $\alpha_{\text{Intelligence}} = -0.25 -0.27$) but increase the probability of UKBB participation ($\alpha_{\text{Income}} = 0.54$, $\alpha_{\text{Education}} = 0.59$, $\alpha_{\text{Intelligence}} = 0.32$)

R1.2. The genetic correlations of reporting error between traits were generally very low, indicating that misreporting one trait, such as birthweight, would have minimal correlation with misreporting other traits. This is not only counter-intuitive but also contradictory to the assumption of the subsequent PCA and the authors' conclusion regarding participants who tend to report more accurately. In their conclusion, the authors suggest that "young, female participants with higher intelligence scores and those from a socio-economic favorable background (higher education and income) tended to provide the most accurate self-report information" (on the first page of the discussion). However, they have shown that there is no systematic reporting error across all phenotypes. The low correlations may be attributed, in part, to the data quality issue mentioned in my previous comment. The authors may also need to reconsider how to interpret the low correlation of repeatability between traits.

With respect to the low correlations between the individual reporting error scores (assuming the reviewer is referring to the phenotypic correlations as shown in Figure 2), it is important to highlight that these scores are only imperfect approximations of their true underlying values, such that attenuated correlation coefficients among those scores are expected. To assess if outlying values among the reporting error scores resulted in attenuation of these correlations, we have now removed the outliers listed above and re-generated the reporting error scores. Of note, since the follow-up duration per individual was the same across most individual reporting error scores for a particular individual, we no longer residualized the individual reporting error scores for follow up duration (to retain the scaling of the scores at this stage) and, instead, residualized the resulting summary score for the follow up duration. Overall, the correction of the reporting error scores did not lead to a noticeable increase in the correlation coefficients or the variance explained by the first principal component (i.e., from 21.05% to 21.06%, c.f., below in the revised Figure 2).

With regard to the reviewer's comment concerning the use of Principal Component Analysis, here our aim was to derive a composite measure that maximises the explained variance among the reporting error scores. While the variance explained by the first principal component obtained from PCA was indeed low (21%), possibly reflecting the imprecision among the individual reporting error scores, 21% explained variance for a loading vector of only positive values is, however, extremely unlikely to occur under the null (when these traits are not correlated). To prove this, we simulated Wishart distributed 5x5 covariance matrices assuming uncorrelated (Gaussian) data for 5 traits in 73,000 samples, computed the resulting correlation matrix (\mathbf{C}) and simulated 100,000 random vectors (\mathbf{v}) with loadings drawn from a uniform distribution on the [0,1] interval. Next, we selected the highest among the computed corresponding $100,000 (\mathbf{v}'\mathbf{C}\mathbf{v})/(\mathbf{v}'\mathbf{v})$ explained variance values. We repeated this procedure 10,000 times and never got the highest eigenvalue corresponding to any positive PC loading as high as the observed 21%. In line with this observation, our PCA-derived reporting error summary score proved useful for the estimation of the construct we were interest in (i.e., reporting error propensity). For example, we have now expanded Figure 2 to provide the correlations between the generated summary score and all individual reporting error scores, including those not used to derive the summary score. As shown, the reporting error summary score showed positive and significant correlation with 23 out of the 33 individual reporting error scores, suggesting that the summary score captures information consistent with an underlying reporting error propensity, despite being only an imperfect approximation of such.

Figure 2. Reporting error summary score

Panel A. Illustration of a reporting error score for a particular phenotype, derived as the residual scores from a model regressing the phenotype measured at time point 2 (P_{T2} , e.g., birth weight reported at follow-up) onto the phenotype assessed at time point 1 (P_{T1} , e.g., self-reported birth weight assessed at baseline). The reporting (residual) error scores are shown as the vertical deviations of the observed values (y_i) around the fitted line. **Panel B.** Correlation matrix highlighting significant ($p < 0.05$) Pearson correlation coefficients between the reporting error scores. Labels in bold highlight variables that were included in Principal Component Analysis. The label highlighted in turquoise ('reporting error summary score') shows the correlations between the PCA-generated summary score (residualized for follow-up time) and the individual reporting error scores. **Panel C.** Summary of results from Principal Component Analysis, highlighting the variance explained by the first principal component (PC1) and the loadings of the indicators on PC1.

To provide the reader with some consideration on our generated composite scores, we have now added the following in the discussion of the manuscript:

[Manuscript, Discussion]: "A key consideration when interpreting our results relates to the error structure examined here. More specifically, our work focused on inconsistent self-reporting over time (i.e., random fluctuations in the phenotype), rather than sources of consistent misreporting (i.e., systematic over- or underreporting, cf. **sFigure 1D**, Supplement). Systematic error, documented for numerous traits (e.g., self-reported weight, where overweight individuals tend to underreport³⁰), can only be explored if error-free reference data is available. For that reason, it was also not possible to explore error in phenotypes subject to temporal variability (e.g., self-reported alcohol use), as the data at hand did not allow us to distinguish reporting error from environmental influences on the observed within-individual variability. In addition, our derived reporting error composite score reflects only an imperfect approximation of its underlying construct (reporting error propensity). Implementing strategies to enhance the resolution of this measure (e.g., by using additional follow-up waves when deriving the individual reporting error scores), alongside explorations of alternative structural models (e.g., single-trait versus single-factor versus multi-factor analyses to capture dimensions of reporting error) could therefore prove useful in future investigations."

Comment 1.3. There were some typos throughout the document. For example, on the first page of the introduction, reference 34 should be 3 and 4.

We have now thoroughly checked and corrected the manuscript for any remaining typos.

Reviewer 2

Given the attention and priority given to the large databanks such as UKBB that is this paper's focus the analyses presented in this carefully laid out paper are of considerable importance. There is too little attention to basic principles of epidemiological research such as reliability and validity. This paper is a refreshing deep dive into this area, and its findings are important. The authors lay out these results and suggest appropriate caution, and potential solutions for the challenges they have surfaced. They rightly highlight that it is not only self report that is subject to error, but the paper focuses on this aspect of UKB, using what should be time invariant variables across time points. In general the paper is written very well and the authors have tried to help the reader understand the sequences of complex analyses and lead them through these in a way that minimises the chances of getting lost. The finding that application of tools designed to increase representativeness of the data may increase bias related to Self report measures is very important. The presentation of findings in relation to the nature of biases at different analytical stages is very useful.

We thank the reviewer for highlighting the importance of our work.

There are some suggestions for further improvement of the manuscript as follows:

Comment 2.1. There should be a clear description (I don't think I missed it) of who is in which analyses, and what are the characteristics of those who are not. The numbers mentioned are not entirely clear- 500,000 goes to around 70,000. It seems that only T1 and T2 are used, rather than looking at many fluctuations- if this is wrong then the sequence of interview timings needs to be laid out more clearly.

Indeed, in our analysis included were only individuals taking part in longitudinal research, where we selected all individuals with complete baseline data and complete data in one follow up assessment. This subset of the UKBB comprises primarily individuals re-invited as part of the first brain magnetic resonance imaging assessment, which aims to re-assess up to 100,000 of the original UKBB volunteers (c.f., reference¹ below). Following the reviewers' suggestion, we have now clarified in the text which individuals were excluded from our analysis, and added a supplement table to show how these individuals differ from those included in this work.

[Manuscript, Methods]: "The UK Biobank is a large prospective study assessing more than 500,000 participants aged between 40 and 69 years who attended one of the baseline assessment centres between 2006 and 2010¹⁷. Included in work were individuals with at least one follow-up assessment, including either individuals taking part in the first repeat assessment centre (around 20,000 participants living within 35 km of the Stockport Biobank coordinating centre¹), or the brain magnetic resonance imaging assessment (ongoing, inviting back up to 100,000 of the original volunteers²)."

References:

- ¹ Lyall, D.M., Cullen, B., Allerhand, M., Smith, D.J., Mackay, D., Evans, J., Anderson, J., Fawns-Ritchie, C., McIntosh, A.M., Deary, I.J. and Pell, J.P., 2016. Cognitive test scores in UK Biobank: data reduction in 480,416 participants and longitudinal stability in 20,346 participants. *PLoS one*, 11(4), p.e0154222.
- ² Miller, K.L., Alfaro-Almagro, F., Bangerter, N.K., Thomas, D.L., Yacoub, E., Xu, J., Bartsch, A.J., Jbabdi, S., Sotiropoulos, S.N., Andersson, J.L. and Griffanti, L., 2016. Multimodal population brain imaging in the UK Biobank prospective epidemiological study. *Nature neuroscience*, 19(11), pp.1523-1536.

[Manuscript, Results]: "Based on PC_1 , we computed the reporting error summary score (RE_{SUM}), which could be generated for 73,127 individuals taking part in repeat assessments. The demographic characteristics of individuals with and without available reporting error summary scores are shown in **sTable 3** (Supplement)."

sTable 3. Demographic and lifestyle characteristics of individuals with and without reporting error scores

Demographic factor	Mean (SD) / %, included sample (n=73127)	Mean (SD) / %, excluded sample (429242)	OR (95% Confidence Interval)	p-value(OR)
Sex (male gender)	48.23%	45.15%	1.13 (1.11; 1.15)	6.162e-54
Age	55.45 (7.54)	56.71 (8.17)	0.98 (0.98; 0.98)	0
Education (length)	17.78 (1.55)	17.17 (1.72)	1.25 (1.25; 1.26)	0
BMI	26.72 (4.34)	27.55 (4.87)	0.96 (0.96; 0.96)	0
Smoking status	0.47 (0.61)	0.57 (0.69)	0.78 (0.77; 0.79)	0
Vigorous physical activity	0.96 (1.38)	0.98 (1.4)	0.99 (0.98; 1)	0.004527
Fluid intelligence score	6.52 (2.05)	5.97 (2.14)	1.13 (1.12; 1.13)	0

Note. OR=Odds ratios estimated for demographic factors predicting missingness in reporting error scores (1=non-missing, 0=missing).

Comment 2.1. The variables of BMI and LDL are treated for these analyses as error free (again if I've understood correctly) but they do have error too. There is a mention of gold standards- however all these types of data have challenges- to see medical records or a single time point 'precision phenotyping' as some sort of gold standard is also not correct, as anyone who has looked at these in detail will have experienced (reliability, validity- what is validity really for some of these variables?). These are all messy attempts at some underlying 'truth', but with very different errors and biases.

We fully agree with the reviewer that objectively ascertained, so called 'gold-standard' measures, are by no means error-free. What we wanted to express is that these measures are (by definition) free of self-report error, although other sources of error will contribute to the observed within-subject variation in these measures. To make this clearer to the reader, we have now added the following:

[Manuscript, Results]: “**Figure 1** also illustrates the level of repeatability for variables containing error due to misreporting and/or temporal variability. Here, self-report measures subject to temporal instability showed particularly low levels of repeatability, notably diet (e.g., vitamin D intake in last 24 hours) and other lifestyles (e.g., physical activity in last 24 hours). While variations among objectively ascertained 'gold-standard' measures (e.g., height, systolic blood pressure, highlighted in violet in **Figure 1**) are free of error due to misreporting, measurement imprecision resulting from other sources (e.g., biological fluctuations, technical challenges, data processing errors) was nevertheless found to be non-negligible for a majority of these measures (e.g., sodium concentration, hearing performance).

Comment 2.2. The use of pre-existing participation bias findings is not explained for readers who are not familiar with the earlier work. Some further description would help, including how robust this work is.

We fully agree that a more detailed description of the existing work focusing on participation bias correction would help the reader interpret the results. We have now added the following in the main manuscript and supplement:

[Manuscript, Methods]: “To explore patterns of covariation between reporting error other participatory behaviours that are known to bias genome-wide estimates, we also included 'UKBB participation probabilities' in the analytical pipeline described above. This trait was derived as part of a previous study¹⁰ focusing on the impact of participation bias on genome-wide findings. The participation probabilities are the predicted probabilities of UKBB participation (with 1= individuals taking part in the UKBB and 0=individuals taking part in a representative reference sample, the Health Survey England¹¹), based on 14 harmonized demographic, social and lifestyle variables. In brief, taking the inverse of the participation probabilities serves as a statistical tool to correct for bias induced by

selective participation, as commonly employed in surveys^{12,13}, classical epidemiological studies^{14,15}, electronic health record studies^{16,17} and, more recently, in volunteer biobank samples^{10,18}. The probability weights included in this work have previously been validated¹⁰, based on external data drawn from representative samples (the Health Survey England¹¹, UK Census Microdata¹⁹) and negative control analyses (genetic analyses on sex²⁰). A more detailed summary of the validation procedures is included in the sMethods (Supplement).

References:

¹¹ Mindell, J., Biddulph, J.P., Hirani, V., Stamatakis, E., Craig, R., Nunn, S. and Shelton, N., 2012. Cohort profile: the health survey for England. *International journal of epidemiology*, 41(6), pp.1585-1593.

¹² Jensen, H.A.R., Lau, C.J., Davidsen, M., Feveile, H.B., Christensen, A.I. and Ekholm, O., 2022. The impact of non-response weighting in health surveys for estimates on primary health care utilization. *European Journal of Public Health*, 32(3), pp.450-455.

¹³ Franco, A., Malhotra, N., Simonovits, G. and Zigerell, L.J., 2017. Developing standards for post-hoc weighting in population-based survey experiments. *Journal of Experimental Political Science*, 4(2), pp.161-172.

¹⁴ Kapteyn, A., Michaud, P.C., Smith, J.P. and Van Soest, A., 2006. Effects of attrition and non-response in the Health and Retirement Study.

¹⁵ Plewis, I., 2007. Non-response in a birth cohort study: the case of the Millennium Cohort Study. *International Journal of Social Research Methodology*, 10(5), pp.325-334.

¹⁶ Beesley, L.J. and Mukherjee, B., 2022. Statistical inference for association studies using electronic health records: handling both selection bias and outcome misclassification. *Biometrics*, 78(1), pp.214-226.

¹⁷ Beesley, Lauren J., and Bhramar Mukherjee. "Case studies in bias reduction and inference for electronic health record data with selection bias and phenotype misclassification." *Statistics in Medicine* 41, no. 28 (2022): 5501-5516.

¹⁸ Salvatore, M., Kundu, R., Shi, X., Friese, C.R., Lee, S., Fritsche, L.G., Mondul, A.M., Hanauer, D., Pearce, C.L. and Mukherjee, B., 2024. To weight or not to weight? The effect of selection bias in 3 large electronic health record-linked biobanks and recommendations for practice. *Journal of the American Medical Informatics Association*, p.ocae098.

¹⁹ 2011 Census Microdata (Office for National Statistics, 2011)

<https://www.ons.gov.uk/census/2011census/2011censusdata/censumicrodata>

²⁰ Pirastu, N., Cordioli, M., Nandakumar, P., Mignogna, G., Abdellaoui, A., Hollis, B., Kanai, M., Rajagopal, V.M., Parolo, P.D.B., Baya, N. and Carey, C.E., 2021. Genetic analyses identify widespread sex-differential participation bias. *Nature Genetics*, 53(5), pp.663-671.

[Supplement, Methods]: The probability weights included in this work were obtained from a previous study focusing on participation bias correction in the UK Biobank¹. A number of robustness checks were implemented to assess the performance of the probability weights, including (1) validation work using the Health Survey England² and the UK Census Microdata³ (n=22,646 and n= 895,649, respectively) and (2) negative control analyses via weighted genome-wide association analysis on sex. With respect to (1), re-weighting UK Biobank participant recovered phenotypic associations as estimated in two representative UK sample. To illustrate, in the (unweighted) UK Biobank sample, there was no phenotypic correlation between age and overall health (r=-0.01), while the observed correlations are in the expected direction in the UK Census Microdata (r=-0.17) and Health Survey England (r=-0.13). Applying the UKBB probability weights recovered these observed correlations (r=-0.13). In addition (2), previous research⁴ has shown that autosomal heritability linked to biological sex could result from sex-differential participation. Comparing the (uncorrected) genome-wide summary statistic results on sex (>2,400,000 participants) to those obtained from weighted genome-wide analyses showed that the application probability weighting reduced artifactual sex-heritability and SNP

effects, providing evidence of diminished (sex-associated) participation bias when increasing sample representativeness.

References

¹ Schoeler, T., Speed, D., Porcu, E., Pirastu, N., Pingault, J.B. and Kutalik, Z., 2023. Participation bias in the UK Biobank distorts genetic associations and downstream analyses. *Nature Human Behaviour*, 7(7), pp.1216-1227.

² Mindell, J., Biddulph, J.P., Hirani, V., Stamatakis, E., Craig, R., Nunn, S. and Shelton, N., 2012. Cohort profile: the health survey for England. *International journal of epidemiology*, 41(6), pp.1585-1593.

³ 2011 Census Microdata (Office for National Statistics, 2011)

⁴ Pirastu, N., Cordioli, M., Nandakumar, P., Mignogna, G., Abdellaoui, A., Hollis, B., Kanai, M., Rajagopal, V.M., Parolo, P.D.B., Baya, N. and Carey, C.E., 2021. Genetic analyses identify widespread sex-differential participation bias. *Nature Genetics*, 53(5), pp.663-671.

Comment 2.3. Exclusion of individuals with high missing variables and (or?) high autosomal heterozygosity needs to be better explained.

We have now expanded the description as follows:

[Manuscript, Methods]: “For all genome-wide analyses (GWA), we restricted the sample to individuals of European ancestry based on principal components and excluded individuals with high missing rate (i.e., proportion of genotypes not called) and/or high heterozygosity on autosomes (i.e., proportion of autosomal heterozygous calls). Here, the UKBB^{1,2} flagged 968 samples outliers due to high missingness and/or extreme heterozygosity that was not explained by mixed ancestry or increased levels of marriage between close relatives. Extreme values in these metrics can be indicators of poor sample quality (e.g., due to DNA contamination) and were therefore discarded.”

¹ Biobank, U.K., 2015. Genotyping and quality control of UK Biobank, a large-scale, extensively phenotyped prospective resource. Available at biobank.ctsu.ox.ac.uk/crystal/docs/genotyping_qc.pdf. Accessed April, 1, p.2016.

² Bycroft C, Freeman C, Petkova D, Band G, Elliott LT, Sharp K, Motyer A, Vukcevic D, Delaneau O, O’Connell J, Cortes A. The UK Biobank resource with deep phenotyping and genomic data. *Nature*. 2018 Oct;562(7726):203-9.

Comment 2.4. Figure 5 needs more explanation.

We have amended Figure 5 and provided more details in the legend to facilitate the reading of that figure:

“Directed Acyclic Graphs (DAGs) illustrating the different simulation settings, including either the ground truth scenario (no participation bias or reporting error, **A1/B1** highlighted in **blue**) or scenarios where reporting error (**A2/B2**, highlighted in **violet**), participation bias (**A3/B3**, highlighted in **green**) or both (**A4/B4**, highlighted in **orange**) were present when assessing the effect of BMI on self-reported education (top panel) and the effect of self-reported education on BMI (bottom panel). The impact of the two participatory behaviours (reporting error, participation) in each of the simulated scenarios was assessed in terms of bias [panel **C1-C2**, showing the difference between the estimated coefficient (y-axis) and the true estimate of the exposure-outcome association (grey line, where the true causal effect was set to be -0.2)] and root-mean-square error (panel **D1-D2**, showing RMSE on the y-axis, with the grey line indicating RMSE=0) when testing the association between education and BMI.”

Reviewer 3

The authors present a comprehensive analysis of an important issue regarding the validity of biobank data, using the UK Biobank and simulations. There is increasing interest in and recognition of the possibility that design elements and assessments in large-scale biobanks may compromise the interpretation of studies leveraging such biobanks. Prior work has shown that participation bias in volunteer biobanks, a growing resource for genomic and epidemiologic research, can lead to biased effect estimates. Here, the authors address the possibility that inconsistent reporting of phenotypic data may induce additional biases. They capitalize on repeated measurements of presumably time-invariant variables (e.g. childhood history of exposures, year of birth, ethnic background) and show that there is apparent reporting error in each and that reporting error and participation biases confer competing bias on effect estimates. Also of note, reporting error bias can substantially decrease effective sample size for biobank analyses.

We thank the reviewer for the positive feedback and summary of our work.

Comment 3.1. Could the authors comment on how the variability of in the repeatability of time-invariant variables (with some e.g. childhood sunburns showing particularly high levels of putative error) might affect the PCA approach used to identify an overall reporting error score?

Included in the PCA approach were only variables with > 50,000 non-missing observations, which is why the reporting error score for number of childhood sunburns ($n=18,372$, $R^2=0.53$), or other phenotypes with particularly low levels of measurement reliability [e.g., comparative body size at age 10 ($n=27,622$, $R^2=0.47$) or relative age of first facial hair ($n=12,905$, $R^2=0.50$)] were not included in the summary score. Among the included reporting error scores, the level of measurement repeatability was largely comparable, including length of education ($n=50,512$, $R^2=0.78$), age when started wearing glasses ($n=62,847$, $R^2=0.85$), age at first sexual intercourse ($n=65,123$, $R^2=0.86$) and father's age at death ($n=54,447$, $R^2=0.95$). Year of birth showed particularly high levels of repeatability ($n=121,248$, $R^2=0.99$), which may explain why this error score showed the lowest loading among the five indicators on the first principal component.

Comment 3.2. It might be worth highlighting the implications of the MR results for studies (e.g. All of Us) that focus on under-represented groups –ie the finding that higher SES is associated with fewer reporting errors and great participation probability. This would presumably induce even greater bias in results for lower SES (and perhaps other groups), suggesting a trade-off between reducing reporting bias and inclusion of more diverse samples.

This is indeed an interesting point. Here, we wish to highlight that our findings are specific to the UK Biobank and may not directly generalise to other samples, in particular if the recruitment strategy, measurement ascertainment and demographic characteristics are not comparable between the samples. We agree that, based on our findings, self-report problems may be more prominent in studies over-sampled for groups of individuals that tend to provide less accurate self-report information. However, while *All of Us* is enriched for individuals historically underrepresented in biomedical research (e.g., individuals of Black/African American ethnicity), the sample still shows similar participation behaviours as documented for the UKBB, considering that included individuals are more educated (42% versus 36% with higher education), more likely to be female (60% versus 52%) and more likely to have health insurance (92% versus 90%) compared to the US general population¹. In addition, as *All of Us* individuals are substantially younger than UKBB participants (52 versus 57 years of age at baseline¹, respectively), which may also impact self-report data quality, it is not possible to anticipate directionalities of potential biases in the *All of Us* sample without additional empirical work. As such, rather than speculating whether the sampling strategies such as those implemented in *All of*

Us lead to more or less bias, we have highlighted that this question would be an interesting avenue for future research:

[Manuscript, Discussion]: “Finally, the reporting error mechanisms identified in this work may not translate to other cohorts, as differences in recruitment schemes and population characteristics likely impact how error in self-report measures is expressed. Future research exploring how self-report error manifests in samples with different characteristics (e.g., enriched for individuals from disadvantaged backgrounds, with poorer health, or younger samples) is therefore needed, to assess how different recruitment strategies may impact bias resulting from reporting error and/or sample non-representativeness.”

Reference

¹ Zeng, C., Schlueter, D.J., Tran, T.C., Babbar, A., Cassini, T., Bastarache, L.A. and Denny, J.C., 2024. Comparison of phenomic profiles in the All of Us Research Program against the US general population and the UK Biobank. *Journal of the American Medical Informatics Association*, p.ocad260.

Comment 3.3. I found it slightly confusing that, after detailed analyses to derive reporting error adjustments, the analysis of SNP weights simply compared GWAS using a baseline measure of a phenotype to GWAS using the average of the two time points. They demonstrate that using the average of measures results in improved power by reducing standard errors of the phenotype. Does this imply that simply averaging repeated measures can be a reasonable approach to minimizing the effect of reporting error in GWAS? Since we don't know the ground truth when dealing with discordant repeated measurements, averaging (at least for quantitative exposures or phenotypes) is straightforward.

The reviewer's conclusion is correct, in that taking the average across multiple measurement occasions will remove some of the random noise in that measurement. In the UKBB, averaging is already common practise for a number of physical measurements assessed multiple times at baseline, such as FVC/FEV (including 3 separate blows assessed at baseline), systolic blood pressure (including two measures taken a few moments apart) or grip strength (one measurement taken per hand), allowing to researchers to easily obtain mean-corrected baseline measures. While mean-correction is indeed a very straightforward way to reduce random error, such strategy requires repeat assessments, which are only available for a smaller subset of the UKBB for self-report measures (i.e., the prospectively ascertained sample). As such, mean-correction can help to reduce reporting error in self-report data, but at the cost of reduced sample size if repeat-measurements are not available for the sample as a whole. We therefore evaluated reporting error weights as an alternative method applicable to potentially all individuals assessed at baseline, provided that a study obtained quality indexes at baseline that could be used to generate reporting error weights. While both approaches can be useful in reducing error in self-report measures, the choice of method will depend on the data availability of a study, which we have now tried to make clearer in the discussion of the manuscript:

[Manuscript, Discussion]: “For example, the application of reporting error (inverse-variance) weights enhanced phenotype resolution in the UKBB without further compromising the level of representativeness in the UKBB (**Figure 4**). Collecting quality indicators and metrics for phenotype precision for all individuals when assessed at study entry (e.g., use of tools to screen for poor questionnaire responding²⁹) in future biobanks may therefore prove useful to remove some of the noise in the phenotype. Alternatively, researchers may choose to average phenotype scores across multiple measurement occasions if repeat-measurement data is available.”

Comment 3.4. The authors make glancing reference to polygenic prediction, but given the widespread use of biobank data for just this purpose and the downward bias of heritability with reporting error, it might be worth a sentence or two more on the implications of this work for biobank PGS research, including the likely variation in reporting error between discovery and validation datasets used for PGS.

We thank the reviewer for raising this point, which we have now discussed in the manuscript as follows:

[Manuscript, Discussion]: “While increasing the sample size (i.e., reduced sampling error) could compensate for the loss of power, such efforts would not correct for the downward bias in estimates of variance components (e.g., SNP heritability, polygenic prediction) resulting from error in the phenotype. [...] For polygenic scores in particular, attenuation bias due to self-report error can be two-fold: First, high phenotypic error in the discovery sample increases measurement error in polygenic indices, leading to attenuation of their effects when tested in replication samples. Second, this bias is expected to be further amplified if self-report accuracy is low in the replication sample.”

Comment 3.5.

5. Figures:

a. Fig 4c: are the 3 panels meant to mirror those in 4b? ie participation, reporting error, and both? Could clarify in legend.

We have now revised the legend and the labelling of Figure 5 in response to reviewer 2.

b. Fig 5: Given structure of the DAGs, does this imply that controlling for (ie conditioning on) factors correlated with participation or reporting could introduce bias (as they are colliders)?

As shown in Figure (e.g., A3), study participation is indeed a collider, since it is a common consequence of both BMI and education. As a result, selecting individuals willing to part in a study would control for a collider (i.e., participation), leading to biased estimates between BMI and education. Reporting error is not a collider in a classical sense, as only its variance is driven by the two variables (education and BMI), and it is not a factor that a study would normally control for (e.g., by selecting individuals with low reporting error). Instead, reporting error in our simulated examples is heteroskedastic error, inducing error in the self-report measure that is nonconstant and varies across individuals, resulting in bias that is different to collider bias. While the two phenomena are different in nature, the solution to both of them is sample (re)weighting.

Overall, this manuscript provides an innovative series of analyses that have valuable implications for biobank research and can inform the design and interpretation of biobank genomic and epidemiologic analyses.

We thank the reviewer for this conclusion.